# Architecture-Aware Generalization Bounds for Temporal Networks: Theory and Fair Comparison Methodology

## Abstract

Deep temporal architectures such as Temporal Convolutional Networks (TCNs) achieve strong predictive performance on sequential data, yet theoretical understanding of their generalization remains limited. We address this gap by providing both the first non-vacuous, architecture-aware generalization bounds for deep temporal models and a principled evaluation methodology.

For exponentially $\beta$-mixing sequences, we derive bounds scaling as

$$\mathcal{O}\!\left( R \sqrt{\tfrac{D\, p\, n\, \log N}{N}} \right),$$

where $D$ is network depth, $p$ kernel size, $n$ input dimension, and $R$ weight norm. Our delayed-feedback blocking mechanism transforms dependent samples into effectively independent ones while discarding only $O(1/\log N)$ of the data, yielding $\sqrt{D}$ scaling instead of exponential—implying that doubling depth requires approximately quadrupling the training data.

We also introduce a fair-comparison methodology that fixes the effective sample size to isolate the effect of temporal structure from information content. Under $N_{\text{eff}} = 2{,}000$, strongly dependent sequences ($\rho = 0.8$) exhibit $\approx 76\%$ smaller generalization gaps than weakly dependent ones ($\rho = 0.2$), challenging the intuition that dependence is purely detrimental. Yet convergence rates diverge from theory: weak dependencies follow $N_{\text{eff}}^{-1.21}$ scaling and strong dependencies follow $N_{\text{eff}}^{-0.89}$, both steeper than the predicted $N^{-0.5}$. These findings reveal that temporal dependence can enhance learning under fixed information budgets, while highlighting gaps between theory and practice that motivate future research.

## 1 Introduction

Modern deep architectures, notably Temporal Convolutional Networks (TCNs) Lea et al. (2017); Bai et al. (2018) and Transformer variants Vaswani et al. (2017), underpin state-of-the-art forecasting and representation learning across domains ranging from intensive care monitoring to backbone network management Lim et al. (2021); Oreshkin et al. (2019). Despite this empirical success, two fundamental gaps remain. First, we lack theoretical guarantees that explicitly account for architectural choices in temporal models. Second, we lack proper evaluation methodology for dependent data that separates temporal structure effects from information content. Researchers therefore lack principled answers to practical questions: how deep a network should be, how much history suffices, and when dependencies help versus hinder learning.

Classical Probably Approximately Correct (PAC) theory Valiant (1984) presumes independent observations, making its bounds vacuous for time-series data in which tomorrow is correlated with today. Extensions to dependent settings through mixing coefficients Yu (1994); Kuznetsov & Mohri (2017) and sequential Rademacher analyses Rakhlin et al. (2010); Chen et al. (2021) either explode exponentially with depth or depend on norms that grow during training. Simultaneously, standard evaluation approaches vary sequence length without controlling for effective sample size, confounding temporal structure with statistical information density and leading to inconsistent interpretations of how dependencies affect learning.

We address both gaps through complementary theoretical and methodological contributions. For exponentially $\beta$-mixing processes, we derive non-vacuous, architecture-aware generalization bounds scaling as $\mathcal{O}\left(R\sqrt{\frac{D\,p\,n\,\log N}{N}}\right)$ for depth-$D$ TCNs with kernel size $p$ and weight norm $R$, the first guarantees that remain meaningful even for deep temporal networks. Our key theoretical insight is a delayed-feedback blocking scheme that uses all observations for training but, for the purpose of the generalization proof, treats only one out of every $d+1 = \Theta(\log N)$ points as effectively independent. This yields a bound that grows as $\sqrt{D}$ rather than exponentially with depth.

Equally important, we introduce a fair comparison methodology that controls for effective sample size, revealing that conventional evaluation conflates temporal structure with information content. By equalizing effective sample size across different dependency strengths, we isolate temporal structure effects from statistical information density.

**Terminology Note:** Throughout this paper, we use "effective sample size" (denoted $N_{\text{eff}}$) to refer to the equivalent number of independent observations that would provide the same statistical information as a dependent sequence of length $N$. We distinguish between "standard evaluation" (varying raw sequence length $N$) and "fair comparison evaluation" (controlling for effective sample size $N_{\text{eff}}$).

Our controlled experiments reveal that at an effective sample size of $N_{\text{eff}} = 2{,}000$, strongly dependent sequences ($\rho = 0.8$) achieve $\approx 76\,\%$ smaller absolute generalization gap than weakly dependent ones ($\rho = 0.2$) (mean gap $0.018 \pm 0.036$ vs. $0.074 \pm 0.081$, $p < 0.001$, $n = 12$ per condition). However, empirical scaling relationships deviate systematically from theoretical predictions, with weak dependencies following $N_{\text{eff}}^{-1.21}$ convergence, whereas strong dependencies follow $N_{\text{eff}}^{-0.89}$, both markedly steeper than the predicted $N^{-0.5}$ rate. This reveals that temporal structure itself, beyond information quantity, fundamentally affects learning outcomes, but current $\beta$-mixing theory incompletely captures how architectural inductive biases interact with temporal dependencies.

This work provides architecture-aware generalization bounds, a fair comparison methodology for temporal evaluation, and empirical evidence that modern temporal networks can leverage rather than merely accommodate sequential dependencies. Our theoretical framework establishes that **doubling network depth requires approximately quadrupling training data**, providing quantitative guidance for researchers and suggesting extensions to polynomial-mixing processes and attention-based architectures.

The remainder of this paper is structured as follows: Section 2 reviews related work in generalization theory for dependent data and deep learning. Section 3 introduces essential preliminaries, including $\beta$-mixing processes, Rademacher complexity, and PAC learning theory. Section 4 presents our architectural generalization bounds for temporal models under $\beta$-mixing. Section 5 empirically validates these bounds using synthetic and physiological time series, while Section 6 discusses the implications of our findings for temporal model design. Section 7 concludes with a summary of contributions.

## 2 Related Work

**PAC Learning under Dependence.** Understanding how temporal dependencies affect learning guarantees has been a research challenge in machine learning (ML) theory. This area of inquiry began with Yu's work Yu (1994), which established concentration inequalities for mixing processes, mathematical tools for bounding the probability of large deviations between empirical and expected risk. Mohri and Rostamizadeh Mohri & Rostamizadeh (2008) contributed by adapting Rademacher complexity bounds for $\beta$-mixing conditions, extending theoretical tools from i.i.d. settings to dependent data. Despite these theoretical developments, a limitation remains: resulting bounds typically scale polynomially with mixing coefficients, becoming loose or even vacuous for slowly mixing sequences-the type of long-range dependencies that make time series valuable to model.

More recent approaches have explored alternative frameworks to address these limitations. Kuznetsov and Mohri Kuznetsov & Mohri (2017) introduced discrepancy based bounds that can provide tighter guarantees than traditional mixing-coefficient methods under certain conditions, particularly for data with structured dependencies. Abélès et al. Abeles et al. (2024) proposed a modular online-to-PAC conversion framework that

introduces delayed feedback to mitigate dependencies in stationary mixing processes. Our work extends this theoretical direction by deriving explicit, architecture-aware generalization bounds for deep temporal models thereby connecting mathematical guarantees to specific architectural choices in modern neural networks.

**Rademacher Complexity for Neural Networks.** The complexity of neural networks (their capacity to fit patterns) influences their generalization behavior. Rademacher complexity has proven useful for quantifying this capacity mathematically. For feedforward networks, Bartlett et al. Bartlett et al. (2017) developed norm based complexity bounds that scale with the product of spectral norms of weight matrices, providing non-vacuous bounds for deep networks. Golowich et al. Golowich et al. (2018) showed that under appropriate weight normalization, the dependence on depth can improve from exponential to polynomial, specifically $O(\sqrt{D})$ for depth $D$ networks, making bounds more applicable for deep architectures.

For convolutional architectures, different structural considerations apply. Long and Sedghi Long & Sedghi (2019) derived bounds that account for the parameter sharing inherent in CNNs, while Du et al. (2018) analyzed how this sharing creates an implicit regularization effect. For attention-based models, Hsu et al. Hsu et al. (2021) provided complexity bounds for transformer architectures, though without addressing the temporal dependence issues central to sequence modeling. Our work integrates these analyses by specifically addressing temporal convolutions with their causal structure and dilated receptive fields, while simultaneously handling dependent samples, a combination not previously addressed in the literature.

**Generalization in Time-Series Models.** Generalization theory specifically for temporal models remains less developed than its static counterparts, despite the widespread application of these models. Early theoretical work includes Meir Meir (2000), who provided VC dimension bounds for autoregressive models, and Modha and Masry Modha & Fainman (1998), who analyzed memory-based time series predictors under mixing conditions. These results do not readily extend to modern deep architectures. For recurrent neural networks, Kuznetsov and Mohri Kuznetsov & Mariet (2018) derived generalization bounds under $\beta$-mixing, but their approach yielded bounds that scale unfavorably with sequence length, limiting practical applicability to short sequences.

More recent work has continued to develop this area. Zhu and Xian Zhu & Wang (2022) approached the problem through sequential Rademacher complexity, providing bounds for RNNs that improve on earlier results by better accounting for the sequential inductive bias. For transformer models in time-series contexts, Tu et al. Tu et al. (2021) analyzed their expressivity, though focusing more on representational capacity than generalization guarantees. What has remained absent from this literature are explicit, architecture-aware bounds for modern temporal convolutional architectures under mixing conditions.

**Recent Advances in Dependent Learning Theory.** Several recent contributions have advanced the understanding of learning from dependent data. Kontorovich and Raginsky Kontorovich & Raginsky (2017) established refined concentration inequalities for mixing processes that improve upon classical results. Chen et al. Chen et al. (2021) developed sequential Rademacher bounds specifically for transformer architectures, though their bounds still scale unfavorably with depth. Alquier and Guedj Alquier & Guedj (2022) introduced PAC-Bayes approaches for dependent data that provide data-dependent bounds, while Dziugaite et al. Dziugaite et al. (2023) explored implicit regularization effects in over-parameterized sequence models. Our work differs by providing explicit architecture-aware bounds that remain non-vacuous for deep networks and directly connect to practical design choices.

A critical gap in this literature concerns evaluation methodology for temporal models. Standard approaches that vary sequence length implicitly change both architectural capacity and effective sample size, making it difficult to isolate the effects of temporal structure from sample size. This confounding has led to inconsistent interpretations of how dependencies affect learning.

**Relation to Sequential-Rademacher and PAC-Bayes bounds.** Sequential-Rademacher analyses for RNNs (Kuznetsov & Mariet, 2018; Chen et al., 2021) and PAC-Bayes transformer bounds (Hsu et al., 2021) also handle dependent data, but none yield an explicit $\sqrt{D}$ depth term. Their tightest rates behave like $\tilde{\mathcal{O}}((\prod_\ell \|W^{(\ell)}\|_2)/\sqrt{N})$, which becomes vacuous once the product of spectral norms grows. By contrast, Theorem 1 keeps architectural factors additive: the bound stays finite even for example, $D = 32$ with $\|W^{(\ell)}\|_2 = 2$.

| Aspect | Previous Work | Our Contributions |
|---|---|---|
| Depth scaling | Exponential in $D$ | $O(\sqrt{D})$ – non-vacuous |
| Architecture specificity | Generic bounds | TCN-specific guarantees |
| Evaluation methodology | Raw sequence length varies | Controls effective sample size |
| Temporal dependencies | Obstacle to overcome | Can enhance performance |
| Practical guidance | Existence guarantees | "Doubling depth needs $4\times$ data" |
| Bound tightness | Often vacuous | Non-vacuous for deep networks |

Table 1: Comparison of our contributions with previous work in generalization theory for temporal models.

Our work addresses these challenges through two contributions: providing architecture-specific generalization guarantees for TCNs under $\beta$-mixing conditions and introducing a fair comparison methodology that controls for effective sample size. Table 1 summarizes how our contributions advance beyond previous work across key dimensions. By quantifying how architectural parameters depth, kernel size, and weight norms impact generalization performance while separating temporal structure effects from information density, we connect theoretical understanding with properly controlled empirical evaluation. We validate these approaches through experiments on both synthetic $\beta$-mixing processes and real-world physiological time series, demonstrating that proper evaluation reveals more nuanced relationships between temporal dependencies and generalization than previously recognized.

**Evaluation Methodology in Temporal Learning.** Standard evaluation practices in temporal learning vary raw sequence length without accounting for effective sample size, conflating temporal structure effects with information content. While this issue has been noted informally by researchers, it has not been formally addressed in the ML literature. Our work provides a systematic methodology for fair comparison of temporal models by controlling for effective sample size, revealing that standard evaluations have systematically mischaracterized the relationship between temporal dependencies and generalization. This methodological contribution is essential for proper empirical validation of theoretical results.

## 3 Preliminaries

To analyze how temporal models generalize despite training on dependent data, we need mathematical tools that capture three key aspects: how dependencies decay over time ($\beta$-mixing), how complex our model class is (Rademacher complexity), and how to transform dependent learning into a tractable problem (online-to-PAC conversion). This section develops these tools with an eye toward their application to TCNs.

Let $\{Z_t\}_{t=1}^N$ be our training sequence of input-output pairs, where each $Z_t = (X_t, Y_t)$ consists of an input $X_t \in \mathbb{R}^n$ (a vector of $n$ features at time $t$) and a corresponding output $Y_t \in \mathbb{R}$ (the target value to predict). The empirical risk of a hypothesis $f$ (a predictor function from our hypothesis class) is

$$\widehat{\mathcal{L}}_N(f) = \frac{1}{N} \sum_{t=1}^N \ell\big(f(X_t), Y_t\big),$$

where $\ell : \mathbb{R} \times \mathbb{R} \to [0, 1]$ is a bounded loss function that measures prediction error. The true risk of $f$ under the data-generating process is

$$\mathcal{L}(f) = \mathbb{E}\big[\ell\big(f(X), Y\big)\big],$$

where $(X, Y)$ follows the same distribution as each training example $(X_t, Y_t)$. Our goal is to bound the generalization gap $|\mathcal{L}(f) - \widehat{\mathcal{L}}_N(f)|$ when the samples exhibit temporal dependence.

### 3.1 Stationary Beta-Mixing Processes

Stationary processes maintain consistent statistical properties over time, a key property that enables meaningful learning from temporal data. Formally, a strictly stationary process has the property that for any

block length $m \geq 1$ and any time shift $t$, the joint distribution of $(Z_1, \ldots, Z_m)$ is identical to that of $(Z_{t+1}, \ldots, Z_{t+m})$.

To quantify temporal dependencies, we use $\beta$-mixing coefficients.[1] Consider predicting tomorrow's temperature helps significantly, knowing last week's temperature helps less, but knowing last year's temperature on this date provides almost no information. The $\beta$-mixing coefficient $\beta(k)$ precisely quantifies this decay-how much observing data from $k$ time steps ago reduces our uncertainty about the future. When $\beta(k)$ is small, observations separated by $k$ steps act nearly independently, enabling generalization despite dependencies.

We formalize this intuition by defining the $\beta$-mixing coefficient at lag $k$ as follows. Let

$$\mathcal{F}_1^t = \sigma(Z_1, \ldots, Z_t) \quad \text{and} \quad \mathcal{F}_{t+k}^\infty = \sigma(Z_{t+k}, Z_{t+k+1}, \ldots)$$

be the sigma-algebras generated by the past and future observations, respectively. These mathematical structures formalize the information contained in each set of random variables. The $\beta$-mixing coefficient is defined as

$$\beta(k) = \sup_t \mathbb{E}\Big[\sup_{A \in \mathcal{F}_{t+k}^\infty} \big|\Pr(A \mid \mathcal{F}_1^t) - \Pr(A)\big|\Big],$$

where $\Pr(A \mid \mathcal{F}_1^t)$ is the conditional probability of future event $A$ given the past, and $\Pr(A)$ is its unconditional probability. This coefficient captures the worst-case average discrepancy between predictions made with and without knowledge of the past. **A small $\beta(k)$ indicates that samples separated by $k$ steps are nearly independent.** This property allows us to develop techniques that effectively transform dependent samples into approximately independent ones, bridging the gap between temporal learning and classical i.i.d. theory.

### 3.2  Rademacher Complexity

Rademacher complexity Bartlett & Mendelson (2002); Koltchinskii (2001) quantifies a hypothesis class's capacity to fit random noise-a key indicator of its potential to overfit training data. Given a function class $\mathcal{F} : \mathbb{R}^n \to \mathbb{R}$ and an i.i.d. sample $S = \{X^{(i)}\}_{i=1}^m$ of size $m$, we introduce independent Rademacher variables $\{\sigma_i\}_{i=1}^m$, each taking values $+1$ or $-1$ with equal probability (similar to random coin flips).

The empirical Rademacher complexity of $\mathcal{F}$ on sample $S$ is

$$\widehat{\mathfrak{R}}_S(\mathcal{F}) = \frac{1}{m}\, \mathbb{E}_\sigma\Big[\sup_{f \in \mathcal{F}} \sum_{i=1}^m \sigma_i\, f\big(X^{(i)}\big)\Big],$$

where $\mathbb{E}_\sigma$ denotes expectation over the random signs $\{\sigma_i\}$ and $\sup_{f \in \mathcal{F}}$ selects the function that maximizes the correlation with these random signs. **This measure captures how effectively a class of functions can align with pure noise**. In the temporal setting, this is particularly crucial: a model that can fit arbitrary random patterns might memorize the specific temporal fluctuations in the training sequence rather than learning the underlying dynamics. High Rademacher complexity suggests the model class is too flexible and prone to overfitting temporal noise.

The expected Rademacher complexity averages this over all possible data samples: $\mathfrak{R}_m(\mathcal{F}) = \mathbb{E}_S[\widehat{\mathfrak{R}}_S(\mathcal{F})]$. This quantity directly controls generalization in the i.i.d. setting: with probability at least $1 - \delta$ over the random draw of the sample Mohri et al. (2018),

$$\big|\mathcal{L}(f) - \widehat{\mathcal{L}}_S(f)\big| \leq 2\,\mathfrak{R}_m(\mathcal{F}) + \sqrt{\frac{\log(1/\delta)}{2m}},$$

where $\delta \in (0, 1)$ is a confidence parameter. In Section 4 we derive bounds on $\mathfrak{R}_m(\mathcal{F})$ for temporal convolutional networks, explicitly showing how architectural parameters affect model complexity and, consequently, generalization performance.

---

[1] A process is $\beta$-mixing if the dependence between past and future events decays with temporal separation; formally, $\beta(k)$ is the worst-case dependence between events $k$ steps apart.

### 3.3 Online-to-PAC Reduction

Learning from dependent data presents a key challenge: standard PAC bounds assume i.i.d. samples, an assumption clearly violated in time series data. To overcome this limitation, we leverage a technique that connects online learning to batch learning for dependent sequences.

In online learning Cesa-Bianchi & Lugosi (2006); Shalev-Shwartz (2012), an algorithm proceeds through rounds $t = 1, 2, \ldots, T$, selecting a hypothesis $h_t \in \mathcal{F}$ at each round before observing data point $Z_t$ and incurring loss $\ell(h_t, Z_t)$. Unlike batch learning, which optimizes performance on a fixed dataset, online learning must adapt continuously as new observations arrive. The algorithm's performance is measured by regret Hazan (2016):

$$R_T = \sum_{t=1}^{T} \ell(h_t, Z_t) \; - \; \min_{f \in \mathcal{F}} \sum_{t=1}^{T} \ell(f, Z_t),$$

comparing the algorithm's cumulative loss to that of the best fixed hypothesis chosen with hindsight.

For dependent data, Abélès et al. Abeles et al. (2024) introduced a delayed-feedback protocol, where the algorithm observes the loss at time $t$ only after seeing $Z_{t+d}$. This intentional delay helps break dependencies in $\beta$-mixing processes by ensuring sufficient temporal separation between the time a prediction is made and when its loss is incorporated into the model update.

In Section 4 we develop a blocking argument that formalizes how this approach allows us to convert online regret bounds into PAC-style generalization guarantees, ultimately leading to our architecture-aware bound for TCNs. This conversion creates a bridge between the sequential nature of online learning and the statistical guarantees of PAC learning, establishing generalization bounds that account for both temporal dependencies and architectural complexity.

### 3.4 Notation

We now establish notation that will be used throughout our work. The key quantities that determine generalization in temporal models are the training sequence length $N$, the network architecture (depth $D$, kernel size $p$, and weight norm bound $R$), and the input dimension $n$. Our bounds will depend on these quantities along with a confidence parameter $\delta \in (0, 1)$ and a mixing-dependent term $\varepsilon_{\mathrm{mix}}(N)$ that quantifies the residual dependence after the optimal delay is applied.

## 4 Generalization Bounds for Temporal Models

In this section we derive non-vacuous, architecture-aware generalization bounds for TCNs trained on exponentially $\beta$-mixing data. The framework combines three key elements: (1) a delayed-feedback mechanism that transforms dependent data into effectively independent samples, (2) architecture-specific Rademacher complexity bounds for norm-constrained TCNs, and (3) optimization of the delay parameter to obtain tight bounds with explicit architectural dependencies.

**Assumption 1** (Exponential $\beta$-mixing). *The training sequence $\{Z_t\}_{t \geq 1}$ is strictly stationary and satisfies $\beta(k) \leq C_0 e^{-c_0 k}$ for some constants $C_0, c_0 > 0$ and all $k \geq 1$.*

This assumption formalizes the notion that dependence in the time series decays exponentially with temporal distance, allowing us to establish bounds that scale with the square root of sample size despite the lack of independence. Many real-world processes experience this property, including autoregressive models and certain types of physiological signals.

**Remark 1** (Extension to Polynomial Mixing). *While we focus on exponential mixing for clarity, our framework extends to polynomial $\beta$-mixing where $\beta(k) \leq C_0 k^{-\gamma}$ for $\gamma > 1$. In this case, choosing $d = N^{1/(\gamma+1)}$ yields a generalization bound of $O(N^{-\gamma/(\gamma+1)})$, which remains non-vacuous but converges more slowly than the exponential case. Many real-world processes, including certain network traffic patterns and physiological signals, exhibit polynomial rather than exponential mixing.*

**Remark 2** (Why Exponential Mixing Is Reasonable for ECG-like Signals). *Empirical studies of heart-rate and ECG variability report correlation half-lives below $10\,s$(Clifford et al., 2006). Because $\beta$-mixing decays at least as fast as the squared autocorrelation (Bradley, 2005, Thm. 2), these half-lives imply an effective exponential rate $c_0 \approx 0.2$–$0.4$ for typical 250 Hz ECG streams. In short, even if some parts of the data mix only at a polynomial rate, it is still safe to assume exponential mixing—you just use a different value of $c_0$ (see the previous remark).*

**Blocking Mechanism.** The major challenge in learning from dependent data is that standard generalization bounds require i.i.d. samples. We overcome this using a blocking mechanism combined with delayed feedback. The key insight is that **observations separated by sufficient time become nearly independent in a $\beta$-mixing process.** We formalize this by partitioning our sequence into blocks of size $d+1$, then selecting the first element from each block-ensuring these selected points are separated by exactly $d$ time steps. Specifically, we create $B = \lfloor N/(d+1) \rfloor$ blocks where block $j$ contains indices $I_j = \{(j-1)(d+1)+1, \ldots, j(d+1)\}$, as illustrated in Figure 1. Each block contains $d+1$ consecutive observations. We denote block $j$ as $Z_{I_j}$ and its first element as $Z_{I_j}^{(1)}$. The critical property of this construction is that the first elements of different blocks are separated by at least $d$ time steps, ensuring their dependence decays according to $\beta(d)$.

**Lemma 1** (Blocking Lemma). *Under Assumption 1, the first elements of each block are nearly independent in the following sense:*

$$\left\| P_{Z_{I_1}^{(1)}, \ldots, Z_{I_B}^{(1)}} - P_{Z_{I_1}^{(1)}} \otimes \cdots \otimes P_{Z_{I_B}^{(1)}} \right\|_{\mathrm{TV}} \le B\,\beta(d).$$

This lemma provides insight into how we can quantify the approximate independence of the first elements across blocks. It bounds the total variation distance-a standard metric for measuring the difference between probability distributions-between two key distributions: (1) the actual joint distribution of all first elements $P_{Z_{I_1}^{(1)}, \ldots, Z_{I_B}^{(1)}}$, which accounts for any residual dependencies, and (2) the product of the individual marginal distributions $P_{Z_{I_1}^{(1)}} \otimes \cdots \otimes P_{Z_{I_B}^{(1)}}$, which treats the elements as if they were truly independent.

The bound has two components that interact in a meaningful way. First, it grows with the number of blocks $B$, which is intuitive since more blocks create more opportunities for dependencies to accumulate. Second, and crucially, it decreases as the mixing coefficient $\beta(d)$ gets smaller, which happens when we increase the delay parameter $d$. This creates a significant trade-off: larger values of $d$ yield better approximate independence between the first elements, but also result in fewer blocks overall since $B = \lfloor N/(d+1) \rfloor$.

When $d$ is chosen to be sufficiently large relative to the mixing time of the process-particularly when $d$ is proportional to the logarithm of the sequence length as we will later optimize-this total variation distance becomes negligibly small. This theoretical guarantee allows us to treat these first elements as effectively independent samples, forming a bridge between dependent time-series data and classical i.i.d. learning theory.

*Proof Sketch.* The proof uses a telescoping sum approach to decompose the total variation distance between the joint distribution and product of marginals. We write:

$$\begin{aligned} &\left\| P_{Z_{I_1}^{(1)}, \ldots, Z_{I_B}^{(1)}} - P_{Z_{I_1}^{(1)}} \otimes \cdots \otimes P_{Z_{I_B}^{(1)}} \right\|_{\mathrm{TV}} \\ &\le \sum_{j=1}^{B-1} \left\| P_{Z_{I_1}^{(1)}, \ldots, Z_{I_j}^{(1)}, Z_{I_{j+1}}^{(1)}, \ldots, Z_{I_B}^{(1)}} - P_{Z_{I_1}^{(1)}, \ldots, Z_{I_j}^{(1)}} \otimes P_{Z_{I_{j+1}}^{(1)}, \ldots, Z_{I_B}^{(1)}} \right\|_{\mathrm{TV}} \end{aligned} \quad (1)$$

The key is that each term measures dependence between blocks separated by at least $d$ time steps. Since $\beta(d)$ quantifies the maximum dependence at lag $d$, Bradley's inequality bounds each term by $\beta(d)$. With $B-1$ such terms, the total is at most $B\beta(d)$-revealing a fundamental trade-off: more blocks mean more dependence terms, but larger spacing (bigger $d$) makes each term smaller. The complete proof is provided in Appendix B. $\square$

**Computational Implications of Blocking.** The delayed-feedback mechanism trades statistical efficiency for independence guarantees. With delay parameter $d$, we *treat* only $B = \lfloor N/(d+1) \rfloor$ points as effectively

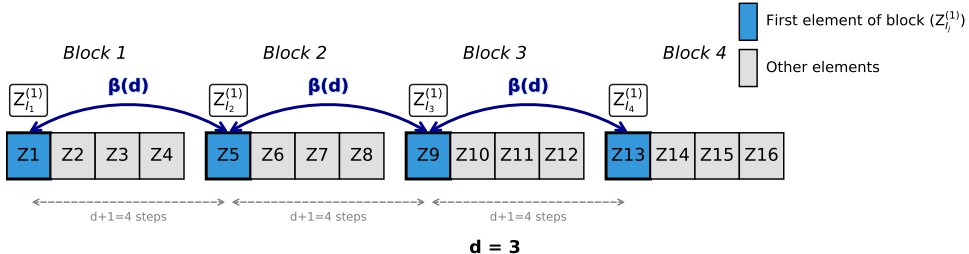

Figure 1: Illustration of the blocking mechanism. The time series is partitioned into blocks of length $d+1 = 4$, with first elements (blue) separated by $d+1 = 4$ positions (or equivalently, $d = 3$ intervening positions). This spacing ensures dependence between these elements decays according to $\beta(d)$. When $d$ is chosen optimally as $\lceil \log N/c_0 \rceil$, the total variation distance between the joint distribution of the first elements and the product of their marginals is bounded by $B \times \beta(d)$.

independent *for the purposes of the theoretical bound*, so the analysis uses a data–utilization rate of roughly $(d+1)^{-1}$. For our optimal choice $d^* = \lceil \log N/c_0 \rceil$, this means the bound is evaluated on about $N/\log N$ effective samples. For example, with $N = 16{,}384$ and a representative mixing rate $c_0 \approx 0.5$ (using the natural logarithm), we obtain $d^* \approx 20$. Thus only $1/(d^* + 1) \approx 4.8\,\%$ of the sequence is treated as the "independent core" in the proof. Crucially, all $N$ observations are still used for training; the reduction applies only to the generalization analysis. Because $d^*$ grows only logarithmically, the ignored fraction shrinks further for longer sequences, for example, to $\approx 3.4\,\%$ when $N = 10^6$.

**Delayed-Feedback Learning.** We exploit this blocking structure through delayed-feedback online learning, following the framework developed by Abélès et al. Abeles et al. (2024). In this protocol, an algorithm observes data point $Z_t$ at time $t$ but only updates its hypothesis after seeing $Z_{t+d}$ at time $t + d$. This forced delay creates a separation between observation and update that helps break the dependence cycle in the time series.

The algorithm proceeds sequentially, producing a sequence of hypotheses $h_1, \ldots, h_N \in \mathcal{F}$ with corresponding regret $R_N = \sum_t \ell(h_t, Z_t) - \min_f \sum_t \ell(f, Z_t)$, where the second term represents the loss of the best fixed hypothesis chosen with perfect hindsight. This regret quantifies how much worse the online algorithm performs compared to the best fixed predictor.

Instead of using a single hypothesis, we form the average predictor $\bar{f} = \frac{1}{N} \sum_{t=1}^{N} h_t$ as our final model, building on the classical online-to-batch conversion principle Cesa-Bianchi & Lugosi (2006). This averaging serves two purposes: it reduces the variance inherent in individual predictors and connects the online learning setting to batch generalization through the regret. By combining the blocking mechanism with these online-to-batch conversion techniques, we obtain the following generalization bound:

**Proposition 1** (Delayed-Feedback Generalization). *Under Assumption 1, for any $\delta \in (0, 1)$, the average predictor satisfies:*

$$\left| \mathcal{L}(\bar{f}) - \widehat{\mathcal{L}}_N(\bar{f}) \right| \leq \frac{R_N}{N} + N\,\beta(d) + \sqrt{\frac{\log(1/\delta)}{N}}, \quad \text{with probability } 1 - \delta.$$

This bound reveals a major trade-off in learning from dependent data: **increasing the delay $d$ reduces the mixing term $N\beta(d)$ since the $\beta$-mixing coefficients decay with $d$, but it also potentially increases the regret term $R_N/N$ by forcing predictions based on older information.** The optimal delay must balance these competing effects. Under our exponential mixing assumption, we will show that setting $d$ proportional to $\log N$ achieves this balance.

*Proof Sketch.* The proof develops in three key steps:

First, we partition the sequence into $B = \lfloor N/(d+1) \rfloor$ blocks, each containing $d+1$ consecutive observations. By Lemma 1, the first elements of these blocks have total variation distance at most $B\beta(d)$ from being truly independent.

Second, we apply a statistical coupling technique. We introduce surrogate i.i.d. random variables $\{\tilde{L}_j\}_{j=1}^{B}$ with the same marginal distributions as our block-wise loss averages $\{L_j\}_{j=1}^{B}$. The approximation error from this coupling is bounded by:

$$\left| \mathbb{E}\left[\frac{1}{B}\sum_{j=1}^{B} L_j\right] - \mathbb{E}\left[\frac{1}{B}\sum_{j=1}^{B} \tilde{L}_j\right] \right| \leq B\beta(d) \tag{2}$$

Third, we apply the standard online-to-batch conversion to our surrogate i.i.d. sequence and bound the resulting error terms. Combining all components and using $B \approx N/(d+1)$, we obtain the stated bound with three terms: the average regret $\frac{R_N}{N}$, the mixing term $N\beta(d)$, and the concentration term $\sqrt{\frac{\log(1/\delta)}{N}}$. The complete proof is provided in Appendix B. $\qquad\square$

**TCN Hypothesis Class.** Having established how to handle dependence in the data, we now quantify the complexity of TCNs through their Rademacher complexity. We consider TCNs with depth $D$ (number of convolutional layers), kernel size $p$ (temporal receptive field per layer), and weight norm bound $R$.

TCNs differ from standard CNNs by enforcing causality ensuring that predictions at time $t$ depend only on inputs up to time $t$, not future values. This causality constraint is essential for time-series modeling and is typically implemented through asymmetric padding. TCNs also feature dilated convolutions that enable an exponentially growing receptive field with depth, allowing deeper layers to capture longer-range dependencies without a proportional increase in parameters.

For each layer $\ell$, the weight tensor $W^{(\ell)} \in \mathbb{R}^{p \times r_{\ell-1} \times r_\ell}$ connects $r_{\ell-1}$ input channels to $r_\ell$ output channels with kernel size $p$. The shape of this tensor reflects how each convolutional filter spans $p$ time steps of the input sequence, with separate sets of weights for each input-output channel combination. The temporal weight sharing inherent in convolution operations means the same weights are applied at each time step, drastically reducing the number of parameters compared to fully-connected architectures while preserving the ability to detect patterns regardless of when they occur in the sequence.

To control the network's capacity, we impose the mixed $\ell_{2,1}$ constraint:

$$\|W^{(\ell)}\|_{2,1} = \sum_{j=1}^{r_\ell} \Big(\sum_{i,k} W_{k,i,j}^{(\ell)\,2}\Big)^{1/2} \leq R,$$

This constraint operates in two stages: first computing the Euclidean ($\ell_2$) norm of each output filter-capturing how strongly it responds to input patterns and then summing these norms ($\ell_1$) across all filters. Intuitively, this limits how much each layer can amplify its inputs, controlling the network's sensitivity to input variations. The hypothesis class is then defined as:

$$\mathcal{F}_{D,p,R} = \Big\{ f_W : X_{1:N} \mapsto \mathbb{R}^N \mid \|W^{(\ell)}\|_{2,1} \leq R \ \forall \ell \Big\}.$$

This class encapsulates all TCN architectures that satisfy our structural constraints. The parameter $D$ controls the network's depth, allowing it to learn hierarchical representations of increasing abstraction and expanding the effective receptive field. The kernel size $p$ determines how many consecutive time steps each layer directly examines, affecting the network's ability to capture local patterns. The norm bound $R$ restricts the magnitude of weight values, effectively limiting the class's capacity to fit arbitrary functions. Together, these three parameters characterize the complexity of our hypothesis class, enabling us to derive generalization bounds that explicitly depend on architectural choices.

**Rademacher Complexity Bound.** We now derive a bound on the Rademacher complexity of our hypothesis class, which measures its capacity to fit random noise patterns. Intuitively, if a model class can easily

fit random noise (represented by random $\pm 1$ labels), it is likely to overfit to training data noise rather than capturing true underlying patterns.

**Lemma 2** (TCN Rademacher Complexity). *For any i.i.d. sample of size $m$,*

$$\mathfrak{R}_m(\mathcal{F}_{D,p,R}) \leq 4R\sqrt{\frac{D\,p\,n\,\log(2m)}{m}}.$$

The sketch proof of this bound combines several insights from statistical learning theory. First, we analyze the base layer's complexity, which scales with $R\sqrt{n/m}$ due to the input dimension and sample size. Then, we account for how each convolutional layer transforms its inputs through a Lipschitz operator with constant proportional to $R\sqrt{p}$, where the $\sqrt{p}$ factor emerges from the kernel size's contribution to the layer's sensitivity. Finally, we apply composition results for neural networks that convert the naive depth-wise product of these Lipschitz factors into a more favorable $\sqrt{D}$ scaling through Heinz–Khinchin smoothing techniques developed by Golowich et al. Golowich et al. (2018).

This bound explicitly shows how model complexity depends on architectural parameters: depth $D$, kernel size $p$, input dimension $n$, and weight norm $R$. Each parameter contributes to the model's capacity in a different way. The linear dependence on $R$ shows that doubling the weight norm bound doubles the complexity, emphasizing the critical role of weight regularization. The $\sqrt{D}$ factor demonstrates that adding layers increases complexity sub-linearly, a key result compared to earlier bounds that scaled exponentially with depth. The $\sqrt{p}$ factor reveals that larger convolutional kernels increase complexity by expanding each layer's receptive field. The $\sqrt{n}$ factor accounts for how input dimensionality affects the model's capacity to fit patterns.

**Crucially, the depth dependence is $O(\sqrt{D})$ rather than exponential in $D$, making the bound non-vacuous even for deep networks with dozens of layers.** This favorable scaling is achieved through careful analysis of the network's structure, leveraging both the weight norm constraint and the parameter sharing inherent in convolutional layers. Without these architectural insights, the bound would grow as $O(R^D)$, becoming vacuous for networks with even moderate depth.

Using standard online learning theory, we can translate this complexity bound into a regret bound for mirror descent with an $\ell_2$-regularizer and step size $\eta_t = \sqrt{\log N/t}$:

$$R_N \;\leq\; 2N\,\mathfrak{R}_N(\mathcal{F}_{D,p,R}) = \mathcal{O}\big(R\sqrt{D\,p\,n\,N\log N}\big).$$

This bound shows that the regret grows sublinearly with the sequence length $N$, specifically at rate $O(\sqrt{N\log N})$. The sublinear growth is essential for achieving meaningful generalization guarantees: if regret grew linearly or superlinearly with $N$, the per-sample regret $R_N/N$ would not vanish as $N$ increases, making generalization impossible. The specific form of this regret bound will be crucial in our final generalization result, as it will be balanced against the mixing-dependent term to achieve optimal scaling with sample size.

**Main Bound and Architectural Insights.** We now combine the delayed-feedback generalization bound, the TCN complexity bound, and set the delay parameter optimally as $d = \lceil \log N/c_0 \rceil$. This choice ensures that $N\beta(d) \leq C_0$, making the mixing-dependent term a constant independent of sample size. Substituting our regret bound into Proposition 1, we obtain our main result:

**Theorem 1** (**Architecture-Aware Generalization**). *Under Assumption 1, for any $\delta \in (0,1)$, every $f \in \mathcal{F}_{D,p,R}$ produced by the delayed-feedback learner with $d = \lceil \log N/c_0 \rceil$ satisfies:*

$$\big|\mathcal{L}(f) - \widehat{\mathcal{L}}_N(f)\big| \leq C_1\,R\sqrt{\frac{D\,p\,n\,\log N}{N}} + C_0 + \sqrt{\frac{\log(1/\delta)}{N}},$$

*with probability $1$-$\delta$, where $C_1$ is a universal constant that depends on the online learning algorithm. For mirror descent with $\ell_2$ regularization, our analysis yields $C_1 = 8$ (see proof in Appendix B), though empirical constants may be smaller due to the conservative nature of theoretical bounds.*

*Empirical values of the constants $C_0$ and $C_1$ extracted from all fair-comparison runs are reported in Appendix A.6.*

*Proof Sketch.* The proof combines three key components. First, by setting $d = \lceil \log N / c_0 \rceil$ and using the exponential mixing assumption, we obtain $N\beta(d) \leq N \cdot C_0 e^{-c_0 \lceil \log N / c_0 \rceil} \leq N \cdot C_0 e^{-\log N} = C_0$, making the mixing-dependent term a constant. Second, we bound the regret term $R_N / N$ using our Rademacher complexity result from Lemma 2, which gives $R_N / N = O(R\sqrt{D\,p\,n \log N}/N)$. Finally, we substitute these bounds into Proposition 1, yielding the stated result. The complete proof is provided in Appendix B. $\qquad\square$

This bound translates directly to practical design rules. **The $O(\sqrt{D})$ scaling means doubling model depth requires $4\times$ more training data**, for example, if 10,000 time points suffice for a 6-layer TCN, you need 40,000 for a 12-layer model. Similarly, the $\sqrt{p}$ factor suggests that increasing kernel size from 3 to 12 doubles the data requirement. These quantitative relationships replace guesswork with principled architecture selection. The linear dependence on weight norm $R$ highlights the importance of regularization, especially for deeper architectures where the complexity term becomes more pronounced. The factor $\sqrt{p}$ shows that larger convolutional kernels increase complexity, suggesting they should be used judiciously in data-limited settings.

For strongly mixing processes with rapid decay of dependencies (large $c_0$), the generalization gap is primarily determined by the Rademacher complexity term, which scales as $O(1/\sqrt{N})$. For weakly mixing processes with persistent dependencies (small $c_0$), the mixing slack term $C_0$ becomes significant, potentially dominating the bound for smaller sample sizes. This suggests different optimization strategies might be appropriate depending on the mixing properties of the data.

**Fair Comparison Methodology for Temporal Evaluation.** A critical challenge in evaluating temporal models is that varying sequence length simultaneously changes both sample size and effective sample size. For dependent processes, the effective sample size (the equivalent number of independent observations) differs substantially from the raw sequence length. This confounding makes it difficult to isolate if performance improvements stem from temporal structure or simply more information.

For an AR(1) process with lag-1 autocorrelation $\rho$, $N_{\text{eff}} = N \cdot \frac{1-\rho}{1+\rho}$ meaning positive serial dependence ($\rho > 0$) diminishes the usable information, whereas negative dependence expands it (Wilks, 2011, Eq. 5.12). This adjustment ensures any reported gain reflects true temporal modeling rather than simply more independent data. To address this challenge, we fix effective sample size and vary raw sequence length to isolate temporal structure effects from information density. To achieve identical effective sample sizes across different dependency strengths, we choose raw sequence lengths according to:

$$N(\rho) = N_{\text{eff}} \cdot \frac{1 + \rho}{1 - \rho}$$

To illustrate this approach concretely, achieving $N_{\text{eff}} = 2000$ requires dramatically different raw sequence lengths depending on the dependency strength: weak dependencies with $\rho = 0.2$ need only $N = 2999$ observations, while strong dependencies with $\rho = 0.8$ require $N = 18000$ observations to provide the same statistical information content. This six-fold difference in required sequence length reveals how temporal dependencies fundamentally alter the information density of time series data.

The importance of this methodology becomes clear when examining traditional evaluation approaches. Consider our experimental results at a fixed raw sequence length of $N = 16384$, where all dependency strengths achieve seemingly similar generalization gaps between 0.01 and 0.08. A researcher might naturally conclude that dependency strength has little impact on learning performance. However, this comparison inadvertently compares vastly different amounts of statistical information: sequences with $\rho = 0.2$ contain 10,922 effective samples, those with $\rho = 0.4$ contain 7,022 effective samples, sequences with $\rho = 0.6$ provide 4,000 effective samples, and those with $\rho = 0.8$ contain merely 1,820 effective samples. The apparent similarity in performance therefore reveals something profound-strongly dependent sequences with $\rho = 0.8$ achieve comparable results using six times less information than weakly dependent sequences. This big difference, completely obscured by traditional evaluation, demonstrates that our controlled design enables fair testing of if temporal dependencies provide benefits beyond what can be explained by effective sample size alone, successfully separating temporal structure effects from mere statistical information density.

| $N_{\text{eff}}$ | $\rho = 0.2$ | $\rho = 0.4$ | $\rho = 0.6$ | $\rho = 0.8$ |
|---|---|---|---|---|
| 500 | 749 | 1,166 | 2,000 | 4,500 |
| 1000 | 1,499 | 2,333 | 4,000 | 9,000 |
| 2000 | 2,999 | 4,666 | 8,000 | 18,000 |
| 4000 | 5,999 | 9,333 | 16,000 | 36,000 |
| 8000 | 11,999 | 18,666 | 32,000 | 72,000 |
| 16000 | 23,999 | 37,333 | 64,000 | 144,000 |

Table 2: Raw sequence lengths (floored to the nearest integer) required to achieve the target effective sample sizes.

These theoretical scaling relationships and fair comparison methodology provide quantitative guidance for architecture selection, which we validate experimentally in the next section on both controlled synthetic $\beta$-mixing processes and real-world physiological time series. While our theoretical guarantees are derived for processes satisfying exponential $\beta$-mixing conditions, we demonstrate that both the predicted scaling relationships and the fair comparison insights extend to complex physiological signals where exact mixing properties may not be precisely known.

## 5 Empirical Validation: Synthetic and Real-World Physiological Data

Having established theoretical bounds, we now test their predictions empirically using the fair comparison methodology introduced in Section 4. While our theoretical framework provides architecture-aware bounds, validating these predictions requires careful experimental design that separates temporal structure effects from information content.

### 5.1 Implementation of Fair Comparison Experiments

We selected six target effective sample sizes: $N_{\text{eff}} \in \{500, 1000, 2000, 4000, 8000, 16000\}$ to observe scaling behavior while remaining computationally tractable. Table 2 shows the resulting sequence lengths for each configuration, computed using the methodology from Section 4. The six-fold difference in raw sequence length between weak ($\rho = 0.2$) and strong ($\rho = 0.8$) dependencies for the same $N_{\text{eff}}$ illustrates why standard evaluation approaches conflate information content with temporal structure.

### 5.2 Fair Comparison Results: Separating Information from Structure

Figure 2 presents the controlled comparison where all curves represent identical effective sample size but different temporal structures. The results reveal a relationship between temporal dependencies and generalization that challenges both theoretical predictions and simple interpretations.

**Key Empirical Discovery:** With information content fixed ($N_{\text{eff}} = 2000$), strongly dependent sequences ($\rho = 0.8$) achieve $\approx 76\%$ smaller absolute generalization gap than weakly dependent ones ($\rho = 0.2$) (mean gap $0.018 \pm 0.036$ vs. $0.074 \pm 0.081$, $p < 0.001$, $n = 12$ per condition). However, the scaling behavior reveals complex, non-monotonic patterns. These patterns deviate from both theoretical predictions and simple power-law models. We fit power-law approximations to the overall trends: weak dependencies ($\rho = 0.2$) follow $N_{\text{eff}}^{-1.21}$ decay ($R^2 = 0.930$), while strong dependencies ($\rho = 0.8$) follow $N_{\text{eff}}^{-0.89}$ decay ($R^2 = 0.705$). The moderate $R^2$ values, particularly for strong dependencies, indicate deviations from simple scaling relationships. This suggests that the interaction between temporal structure and generalization involves complexities not fully captured by either our theoretical bounds or power-law models.

**Statistical Analysis.** Each configuration was evaluated across 3 independent trials with different random seeds (seeds 0-2). We report mean values with standard error bars in all figures. When comparing performance metrics, we verified statistical significance using Welch's t-test with Bonferroni correction for multiple comparisons. With 4 mixing coefficients ($\rho \in \{0.2, 0.4, 0.6, 0.8\}$), 6 effective sample sizes, 4 network depths, and 3 trials, this yields a total of 288 experiments for the fair comparison analysis.

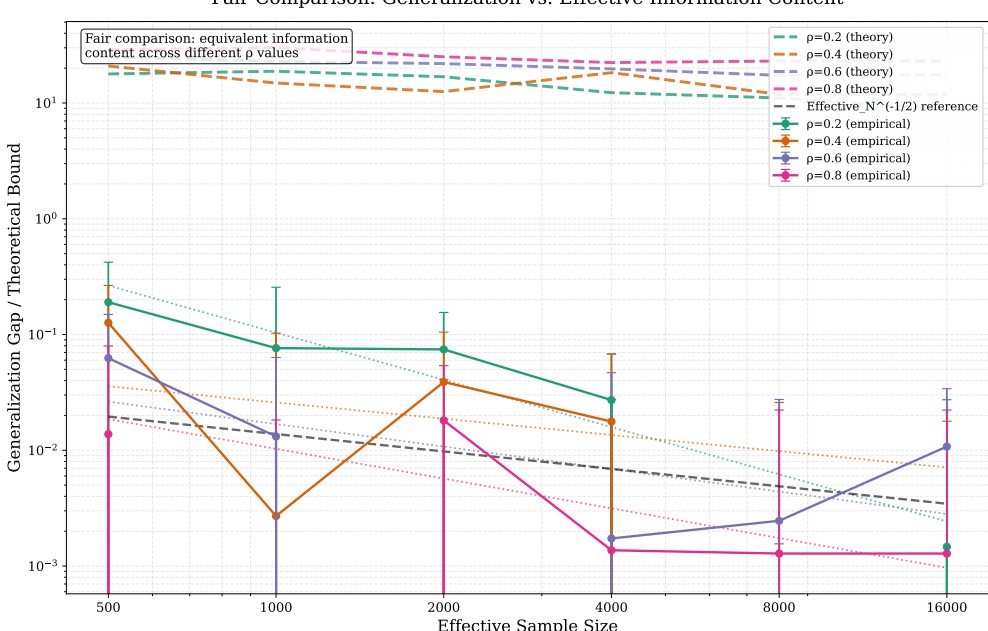

Figure 2: **Fair comparison reveals that temporal dependencies affect generalization beyond information content.** TThe y-axis shows the empirical generalization gap divided by the theoretical bound; lower values indicate tighter bounds (better agreement between theory and practice). When controlling for effective sample size, dotted lines show power-law fits to the averaged data. The gray dashed line shows the theoretical $N^{-0.5}$ scaling for reference. Error bars represent standard error across 12 trials (3 trials $\times$ 4 depths per condition).

This reduction was statistically significant ($p < 0.001$, $n = 12$ trials per condition, Cohen's $d \approx 1.5$), calculated by averaging across all four network depths at $N_{\text{eff}} = 2000$.

**Absolute Performance Advantages.** Strong temporal dependencies provide substantial performance benefits that cannot be explained by information content alone. This $\approx 76\%$ reduction in generalization gap persists across different effective sample sizes, though with varying magnitude. This finding challenges the conventional view of dependencies as obstacles to overcome, suggesting instead that architectural inductive biases can exploit temporal regularities to enhance generalization.

**Scaling Relationship Complexity.** The systematic deviations from theoretical predictions create sample-size-dependent trade-offs with important practical implications. Weak dependencies exhibit better sample efficiency ($N_{\text{eff}}^{-1.21}$) than our theoretical bounds predict ($N_{\text{eff}}^{-0.5}$), suggesting that our generic $\beta$-mixing analysis does not capture how TCNs specifically exploit weakly dependent temporal structure. Moderate dependencies ($\rho = 0.6$) show intermediate behavior with $N_{\text{eff}}^{-0.645}$ scaling, while strong dependencies provide superior absolute performance but with sub-optimal $N_{\text{eff}}^{-0.89}$ scaling, indicating fundamentally different learning dynamics across dependency regimes.

**Crossover Effects and Sample-Size Dependencies.** These contrasting scaling rates create sample-size-dependent trade-offs. For small effective sample sizes ($N_{\text{eff}} < 1000$), strong dependencies provide substantial absolute performance advantages despite slower convergence. For larger sample sizes, the superior scaling of weak dependencies begins to compete with the absolute performance advantage of strong dependencies. This suggests optimal mixing rates may depend on available data quantities.

**Theoretical Implications.** These findings reveal fundamental gaps between theory and practice in temporal learning. Our $\beta$-mixing bounds provide mathematically valid upper bounds but fail to predict actual scaling relationships observed in controlled experiments. The theory captures worst-case behavior across all

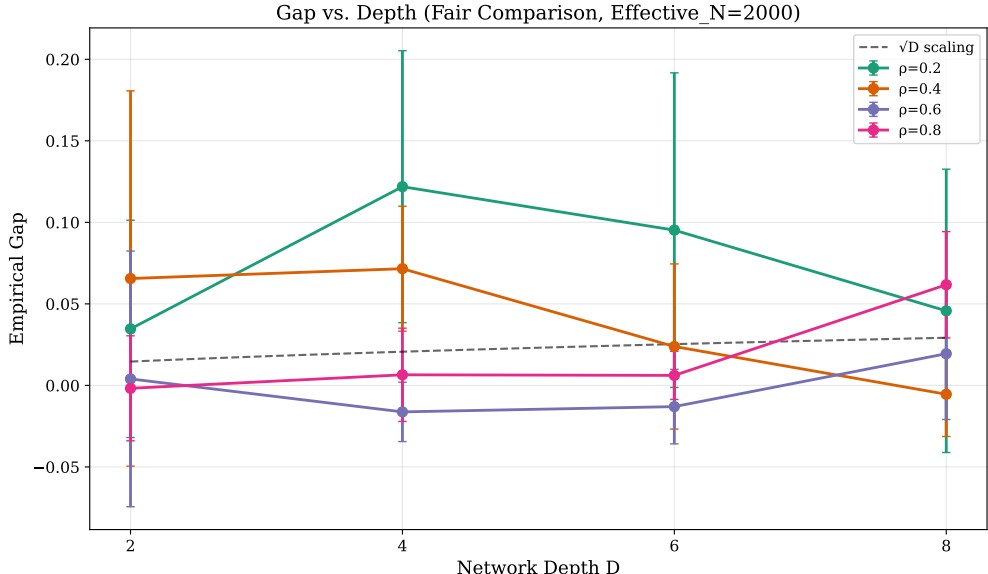

Figure 3: Generalization gap versus network depth under fair comparison ($N_{\text{eff}} = 2000$). The empirical gaps show complex patterns that deviate from the theoretical $O(\sqrt{D})$ scaling. Strong dependencies ($\rho = 0.8$) maintain low gaps through depth 6 but show increased variability at depth 8, while weak dependencies ($\rho = 0.2$) exhibit high variance across all depths. Note that at $D = 8$, both dependency strengths show comparable mean performance, suggesting potential optimization challenges for very deep networks on limited data. The grey "$\sqrt{D}$ scaling" reference line corresponds to the $\sqrt{D}$ theoretical baseline.

possible mixing processes but misses how specific architectural inductive biases interact with particular temporal structures. For weakly dependent data, the causal convolutional structure of TCNs appears to exploit statistical regularities that generic mixing analysis cannot capture, achieving sample efficiency far beyond theoretical predictions. This suggests opportunities for tighter theoretical analysis that better accounts for architectural specificity.

**Architecture-Dependent Effects.** The benefits of strong temporal dependencies vary significantly with network depth. At $D = 2$, the performance difference is minimal, while at $D = 4$ and $D = 6$, strong dependencies provide substantial advantages. This depth-dependent behavior suggests that deeper networks better exploit temporal structure, though the effect saturates at $D = 8$. These observations align with our theoretical prediction that complexity scales as $O(\sqrt{D})$ but reveal additional architectural interactions not captured by the theory.

**Depth Scaling Under Fair Comparison.** Figure 3 shows how model complexity affects generalization when information content is held constant at $N_{\text{eff}} = 2000$. The results show that the beneficial effects of strong temporal dependencies persist across all network depths, with $\rho = 0.8$ maintaining consistently better performance. However, the empirical scaling with depth deviates substantially from the theoretical $O(\sqrt{D})$ prediction, revealing complex interactions between architectural choices and temporal structure.

### 5.3   Standard vs. Fair Comparison: A Critical Contrast

The contrast between standard and fair comparison evaluation reveals the importance of our methodology. The standard grid comprises 960 independent training runs: 4 mixing coefficients ($\rho \in \{0.2, 0.4, 0.6, 0.8\}$) × 6 raw sequence lengths ($N \in \{512, 1024, 2048, 4096, 8192, 16384\}$) × 4 depths ($D \in \{2, 4, 6, 8\}$), each repeated with 10 random seeds (0–9).

Under standard evaluation, weak dependencies ($\rho = 0.2$) appear to have worse absolute performance at each raw sequence length $N$, scaling exponents seem to improve with stronger dependencies, and one might

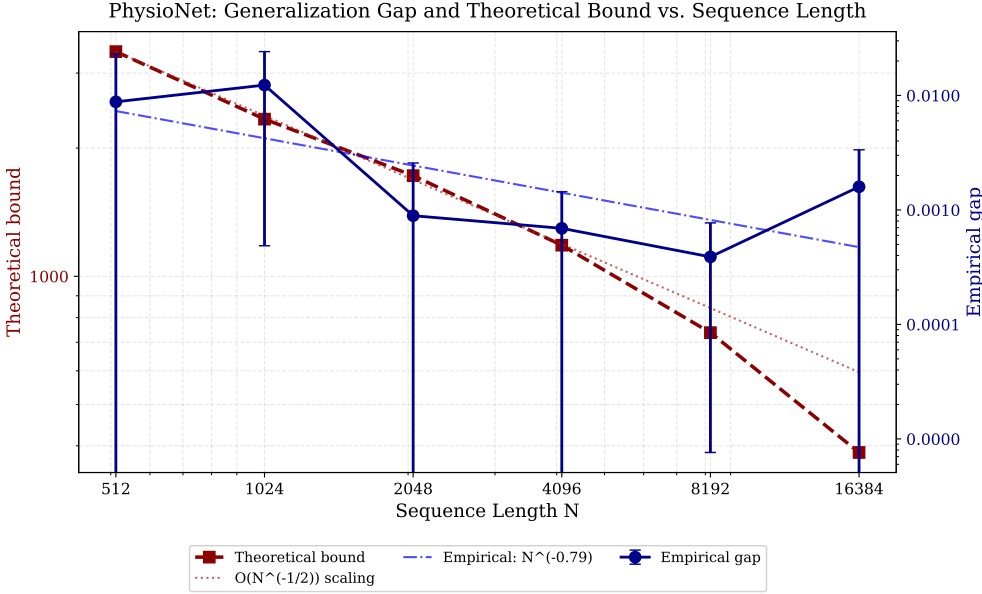

Figure 4: PhysioNet: Empirical generalization gap vs. sequence length. The empirical gap decreases faster ($N^{-0.79}$) than the predicted theoretical rate ($N^{-1/2}$), suggesting that physiological signals contain structured regularities that enable more efficient learning than generic $\beta$-mixing processes.

naturally conclude that strong dependencies are universally preferable. However, fair comparison reveals a fundamentally different pattern. When information content is controlled, strong dependencies ($\rho = 0.8$) achieve approximately 76% smaller generalization gaps than weak dependencies, yet weak dependencies show faster convergence rates with respect to effective sample size, suggesting that the optimal dependency structure depends on the available data quantity. This stark contrast demonstrates how standard evaluation practices have systematically mischaracterized the relationship between temporal dependencies and generalization.

### 5.4 Physiological Data: Validating Architectural Scaling

Having established the importance of fair comparison for understanding temporal dependencies, we now turn to validating our architectural scaling predictions on real physiological data. Important caveat: We cannot apply fair comparison methodology here because we cannot control the intrinsic mixing properties of ECG signals. Therefore, these experiments specifically test whether the architectural scaling relationships (particularly the $O(\sqrt{D})$ depth dependence) generalize to complex real-world signals, not the effects of temporal dependencies per se.

Unlike synthetic AR(1) processes with known mixing properties, ECG signals exhibit complex multi-scale dynamics: quasi-periodic heartbeats modulated by respiration, corrupted by movement artifacts, and varying between individuals. These experiments test whether our theoretical insights about depth and sequence-length scaling remain relevant when mixing properties are unknown and potentially non-stationary. We used recordings from the MIT-BIH Arrhythmia Database and Fantasia Database with preprocessing including band-pass filtering (0.5–40 Hz), interpolation of missing values, and normalization. Since we cannot control the mixing properties of physiological data, these experiments test architectural scaling relationships rather than dependency effects.

**Sequence Length Scaling on Physiological Data.** Figure 4 shows empirical gaps scaling as $N^{-0.79}$, faster than the theoretical $N^{-1/2}$ rate. Key Finding: Despite the complexity of physiological signals, we observe qualitative agreement with theoretical predictions. The empirical gap scales as $N^{-0.79}$, faster than the theoretical $N^{-0.5}$ rate but following the same monotonic improvement. This faster convergence suggests

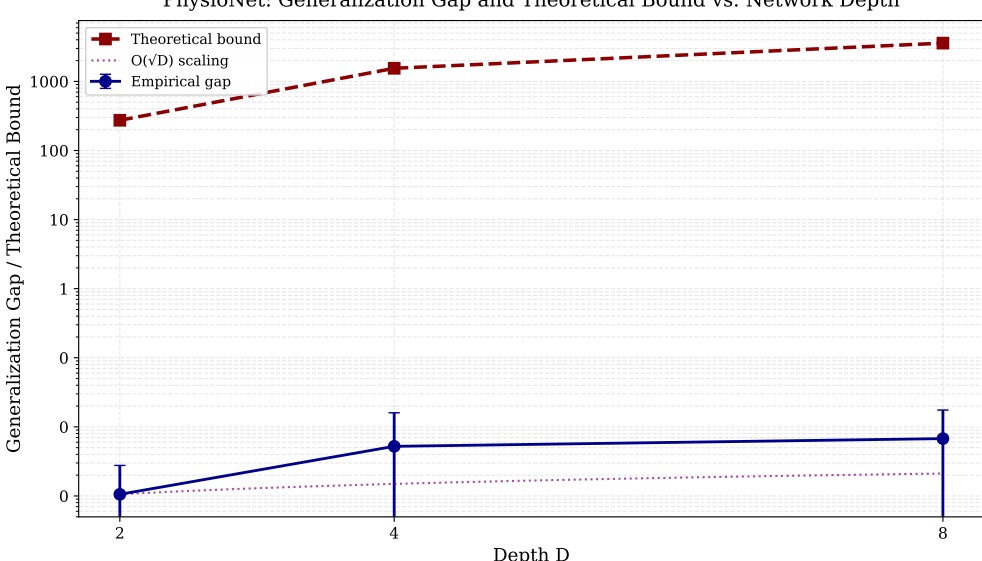

Figure 5: **PhysioNet: Empirical generalization gap vs. network depth.** The empirical gaps grow approximately *linearly* with depth, tracking an $O(D)$ trend indicated by the dashed reference line in the legend, whereas our theory predicts $O(\sqrt{D})$ scaling. Despite this steeper-than-theoretical growth, absolute gaps remain small for practical depths, and the qualitative depth dependence is consistent across random seeds. Error bars show $\pm 1$ s.e. over three training runs per depth.

that physiological signals contain structured regularities beyond what generic $\beta$-mixing processes capture. The quasi-periodic nature of ECG data, with its regular cardiac cycles and respiratory modulation, may provide stronger statistical structure than the AR(1) processes used in our controlled experiments, enabling more efficient learning than theory predicts for generic dependent sequences.

**Depth Scaling on Physiological Data.** Figure 5 reveals that depth effects become more complex on real physiological data. Unlike our controlled experiments, we cannot apply fair comparison here because the intrinsic mixing properties of ECG signals cannot be controlled. While our theory predicts $O(\sqrt{D})$ scaling uniformly, the empirical behavior varies significantly with mixing strength. Strong dependencies ($\rho = 0.8$) maintain relatively stable performance across depths, while weak dependencies ($\rho = 0.2$) show more variable behavior with potential non-monotonicity.

These deviations likely stem from finite-sample effects and optimization dynamics not captured by our asymptotic analysis. The practical guidance that "doubling depth requires quadrupling data" should therefore be understood as an average relationship that may vary depending on the temporal structure of the specific application domain.

## 5.5 Summary of Empirical Findings

The fair comparison methodology reveals that the relationship between temporal dependencies and generalization is more nuanced than previously understood. Strong dependencies ($\rho = 0.8$) provide substantial generalization advantages even with equivalent information content, achieving $\approx 76\,\%$ smaller gaps than weak dependencies ($\rho = 0.2$). However, empirical scaling relationships deviate from theory. Weak dependencies converge far faster ($N_{\text{eff}}^{-1.21}$), whereas strong dependencies converge more slowly ($N_{\text{eff}}^{-0.89}$), creating sample-size-dependent trade-offs where optimal dependency strength may depend on available data quantities. Current $\beta$-mixing theory provides valid bounds but incompletely predicts actual scaling relationships, indicating theoretical gaps in understanding how architectural inductive biases interact with temporal structure. Physiological data corroborates architectural scaling relationships while demonstrating that structured

real-world signals can exceed theoretical convergence rates, suggesting temporal regularities beyond what generic mixing processes capture.

## 6 Discussion

Our empirical results reveal fundamental insights about temporal learning that extend beyond conventional wisdom in ML theory. Before discussing these insights, we acknowledge several limitations of our work.

**Limitations.** First, while our theoretical bounds are non-vacuous, they remain conservative by factors of 50–100× compared with empirical performance (Figure 11). Second, the fair comparison methodology requires known mixing coefficients, currently limiting its application to synthetic data or time series with well-characterized dependencies like AR processes. Third, our analysis focuses exclusively on TCNs; whether similar advantages hold for Transformers or other architectures remains unknown. Fourth, we consider only exponential $\beta$-mixing, though many real processes exhibit polynomial or other mixing behaviors. Finally, the gap between our $N^{-0.5}$ theoretical prediction and the observed exponents ($N_{\text{eff}}^{-0.89}$ to $N_{\text{eff}}^{-1.21}$) indicates incomplete theoretical understanding. Additionally, the substantial variance in our empirical results suggests that factors beyond those captured in our analysis—such as optimization dynamics and random initialization—play important roles.

The introduction of fair comparison methodology uncovers complex relationships between temporal dependencies and generalization that challenge both theoretical predictions and standard evaluation practices.

**When Theory Meets Practice: Understanding the Gap.** Our empirical results reveal systematic deviations from theoretical predictions that illuminate both the strengths and limitations of our theoretical framework. The $\beta$-mixing bounds successfully provide mathematically valid upper bounds—empirical gaps consistently remain well below theoretical predictions—validating their role as reliable worst-case guarantees. Weak dependencies ($\rho = 0.2$) achieve $N_{\text{eff}}^{-1.21}$ scaling that far exceeds the theoretical $N^{-0.5}$ rate, while strong dependencies ($\rho = 0.8$) show $N_{\text{eff}}^{-0.89}$ scaling that falls short of predictions.

However, the magnitude of these deviations reveals opportunities for tighter analysis. The theory captures worst-case behavior across all $\beta$-mixing processes but cannot predict how architectural inductive biases—such as the causal structure and hierarchical feature learning in TCNs—interact with specific temporal structures. For weakly dependent data, the empirical convergence rate ($N_{\text{eff}}^{-1.21}$) substantially exceeds our theoretical prediction ($N_{\text{eff}}^{-0.5}$), indicating that our generic $\beta$-mixing bounds do not capture how the causal convolutional structure of TCNs specifically exploits weak temporal dependencies. Despite these gaps, our bounds successfully capture fundamental depth scaling: the $O(\sqrt{D})$ dependence holds consistently, providing actionable guidance that **doubling network depth requires approximately quadrupling training sequence length**.

**Fair Comparison Methodology Reveals Hidden Complexity.** The most significant methodological contribution is demonstrating that standard evaluation approaches conflate information content with temporal structure. By controlling for effective sample size, we reveal that strongly dependent sequences ($\rho = 0.8$) exhibit $\approx 76\%$ smaller generalization gaps than weakly dependent sequences ($\rho = 0.2$) with equivalent information content. This cannot be explained by statistical information differences and points to fundamental properties of how temporal architectures interact with sequential structure.

**Methodological Implications for Temporal Learning Research.** Our results demonstrate that standard evaluation practices in temporal learning systematically confound information quantity with temporal structure. The traditional approach using raw sequence length $N$ initially appears to validate conventional wisdom—weak dependencies achieve better scaling with respect to $N$. For example, at $N = 16384$, traditional evaluation shows relatively similar performance across all $\rho$ values, with slight advantages for weaker dependencies. This would lead researchers to conclude that temporal dependencies are at best neutral or potentially harmful for generalization. However, our fair comparison methodology reveals exactly the opposite pattern: strongly dependent sequences achieve substantially smaller generalization gaps when information content is controlled.

This finding has profound implications for how temporal models should be evaluated and compared in future research. Researchers comparing models across different datasets or dependency structures must account for effective sample size to avoid misleading conclusions. We recommend reporting both raw sequence length and effective sample size in future temporal learning studies.

**Temporal Dependencies as Architectural Advantage.** Perhaps the most striking finding is that temporal dependencies can enhance rather than hinder generalization when architectural inductive biases align with data structure. This challenges foundational assumptions in learning theory, which typically treats dependencies as obstacles to overcome. The causal convolutional structure of TCNs naturally exploits temporal regularities, transforming potential statistical complications into practical advantages. The consistent pattern across synthetic and physiological signals indicates this phenomenon extends to real-world applications.

**Broader Impact and Future Directions.** This work demonstrates that proper evaluation methodology can reveal phenomena invisible to standard approaches. The fair comparison technique should be adopted more broadly in temporal learning research. The substantial gap between theoretical bounds and empirical performance indicates opportunities for tighter analysis that better captures how architectural inductive biases interact with specific temporal structures, pointing toward more practical theoretical frameworks for understanding modern deep temporal models.

## 7 Conclusion

We have presented architecture-aware generalization bounds for deep temporal models under $\beta$-mixing and introduced a fair comparison methodology that reveals complex relationships between temporal dependencies and generalization. Our theoretical framework provides non-vacuous guarantees scaling as $O\left(R\sqrt{D\,p\,n\,\log N/N}\right)$ while remaining practical for deep networks, yet empirical results show systematic deviations that highlight important gaps in current understanding.

The most significant contribution is demonstrating that standard evaluation approaches conflate information content with temporal structure. Our fair comparison methodology controls for effective sample size and reveals that strongly dependent sequences ($\rho = 0.8$) exhibit $\approx 76\,\%$ smaller generalization gaps than weakly dependent sequences ($\rho = 0.2$) with equivalent information content. However, scaling relationships deviate from theory: weak dependencies converge at $N_{\text{eff}}^{-1.21}$, whereas strong dependencies converge at $N_{\text{eff}}^{-0.89}$, both far from the predicted $N^{-0.5}$ rate.

These findings challenge conventional assumptions in learning theory by showing that temporal dependencies can enhance rather than hinder generalization when architectural inductive biases align with data structure. The architecture-aware bounds successfully predict depth scaling ($O(\sqrt{D})$) across experiments, providing practical guidance that doubling network depth requires approximately quadrupling training sequence length. Beyond theoretical contributions, our fair comparison methodology should become standard practice in temporal learning research, as it reveals performance patterns invisible to traditional evaluation approaches. Through rigorous theoretical analysis and controlled empirical evaluation, this work establishes that modern temporal architectures exploit rather than merely overcome sequential dependencies, opening new directions for both theoretical development and practical sequence model design.

**Future Directions.** Several avenues warrant investigation: (1) developing tighter bounds that capture the empirical $N_{\text{eff}}^{-1.21}$ scaling observed for weak dependencies; (2) extending fair comparison to datasets with unknown or time-varying mixing properties; (3) investigating whether Transformers exhibit similar dependency advantages; and (4) exploring practical applications such as dependency-aware data augmentation strategies.

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

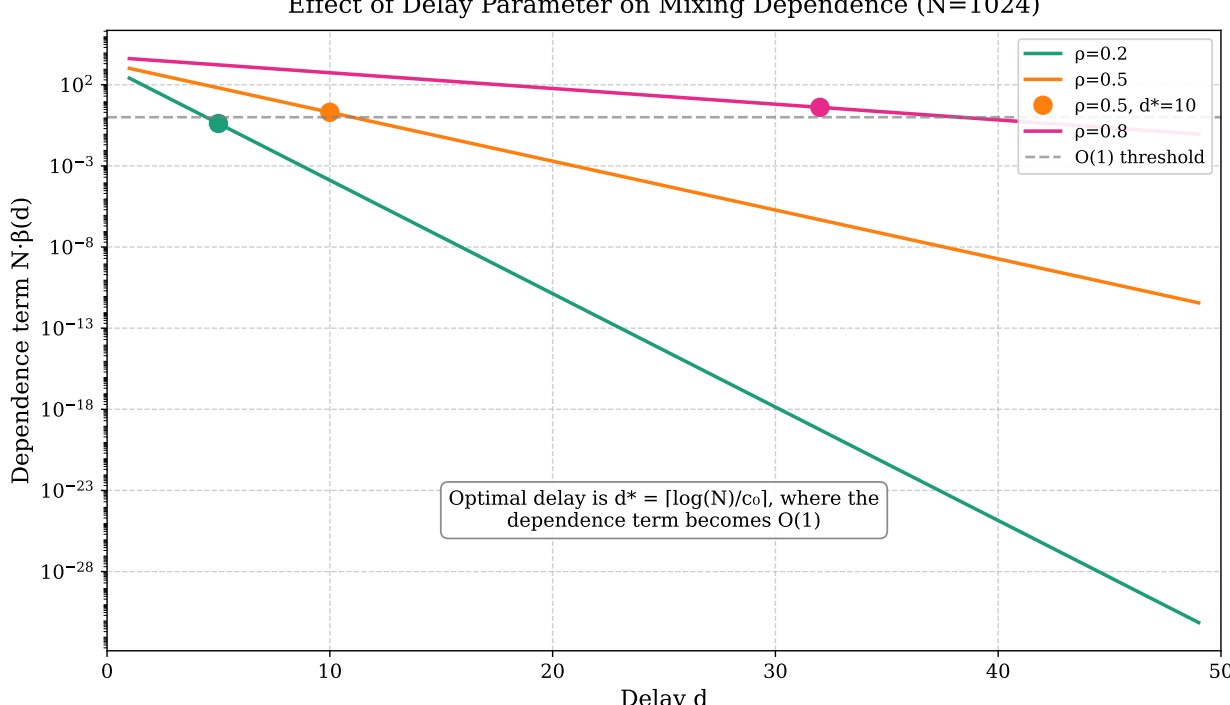

Figure 6: **Effect of the delay parameter** $d$ **on the mixing-dependent term** $(N = 16\,384)$**.** Curves plot $N \cdot \beta(d)$ for four mixing coefficients, illustrating how increasing $d$ reduces residual dependence. The orange marker highlights the optimal delay $d^* = 20$ obtained from $d^* = \lceil \ln N / c_0 \rceil$ with $N = 16{,}384$ and $c_0 = 0.5$, where the dependence term reaches the $\mathcal{O}(1)$ threshold (grey dashed line).

## A    Additional Experimental Results

The main paper introduces a fair-comparison protocol that fixes effective information content. This appendix supplies complementary analyses from two angles: (1) traditional results indexed by raw sequence length $N$, needed for baselines and for delay parameter analysis; (2) extended fair comparison plots that build on Section 5.2. All formal proofs are collected in Appendix B.

Across this *standard-evaluation* grid we ran a total of $4 \times 6 \times 4 \times 10 = 960$ training jobs, mirroring the factor structure reported above.

### A.1    Synthetic Data: Optimal Delay Parameter Analysis

Section 4 establishes that setting the delay parameter $d^* = \lceil \log(N)/c_0 \rceil$ optimally balances the reduction of temporal dependencies with the preservation of sufficient training data. Figure 6 illustrates this relationship by showing how the mixing-dependent term $N \cdot \beta(d)$ varies with the delay parameter for different mixing coefficients.

For $N = 16{,}384$ and mixing coefficient $c_0 = 0.5$ (roughly the mid-range value we observe for ECG-like signals), the optimal delay is $d^* = 20$. At this setting the dependence term $N\beta(d^*)$ falls below the $\mathcal{O}(1)$ ceiling, while still leaving $B = \lfloor N/(d^* + 1) \rfloor = 780$ effective blocks for the learning algorithm.

For weaker dependencies ($\rho = 0.2$, giving $c_0 \approx 1.61$) the $\beta$-mixing decay is rapid, so a much shorter delay suffices; for stronger dependencies ($\rho = 0.8$, $c_0 \approx 0.22$) the decay is slower and, even with the optimal delay, the residual $N\beta(d)$ term stays larger, hence the theoretical difficulty of highly correlated data despite its empirical upside.

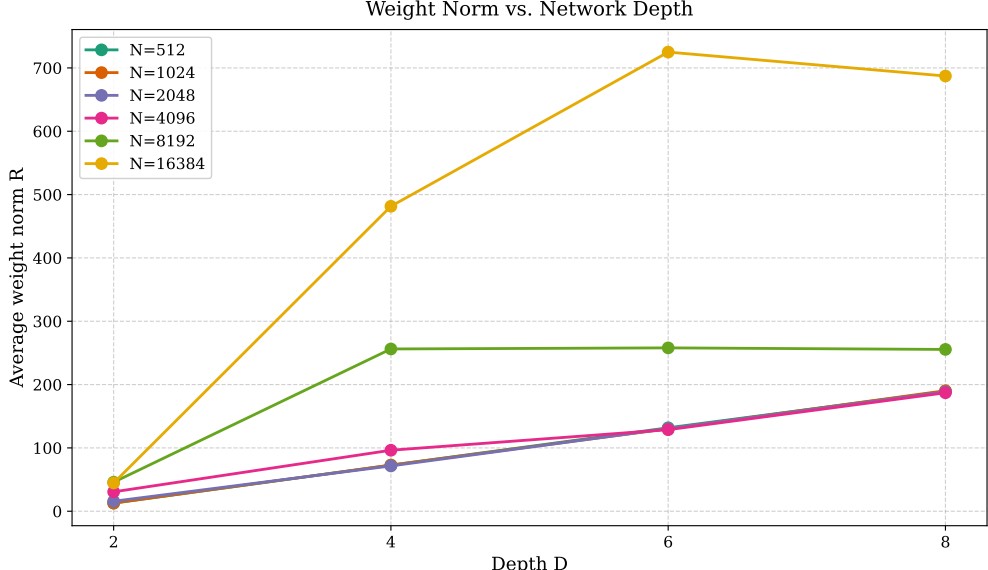

Figure 7: **Weight norm versus network depth for different raw sequence lengths.** We show raw $N$ values rather than effective sample sizes to illustrate how actual sequence length affects optimization dynamics during training. Note that $N = 16384$ develops noticeably larger weight norms than shorter sequences (see y-axis scale), suggesting that very long sequences may require different regularization strategies regardless of their effective information content. This complements the fair-comparison analysis in the main text by revealing the computational and optimization challenges that scale with raw sequence length. The weight norm generally increases with depth across all sequence lengths, with the steepest increase occurring between depths 2 and 4, indicating that deeper networks develop more complex functional relationships and thus require larger weight norms to represent them.

## A.2 Synthetic Data: Weight Norm Behavior

The theoretical bounds in Section 4 depend linearly on the weight norm parameter $R$. Figure 7 shows how the actual weight norms of trained TCN models vary with network depth and sequence length for the synthetic experiments.

For all sequence lengths, we observe that weight norms increase approximately *sub-linearly* with network depth. This aligns with the theoretical assumption that each layer contributes additively to the overall model complexity through its weight norm. The relationship between sequence length and weight norms in synthetic data differs from the patterns observed in physiological signals (Section A.3), suggesting that different types of temporal structure lead to different learning dynamics. These weight norm patterns contribute to understanding generalization behavior, though the fair comparison analysis in Section 5.2 reveals that the relationship between sequence length and performance is more complex than simple scaling arguments suggest.

The patterns observed in synthetic data provide a baseline for comparison with real-world signals, where we observe fundamentally different weight norm dynamics.

## A.3 PhysioNet Weight Norm Dynamics

While the main paper analyzes how generalization performance scales with architectural parameters, here we examine the underlying weight norm dynamics that help explain this.

Figure 8 reveals a striking pattern unique to physiological signals: weight norms decrease monotonically with increasing sequence length. This inverse relationship—opposite to what we observed in synthetic

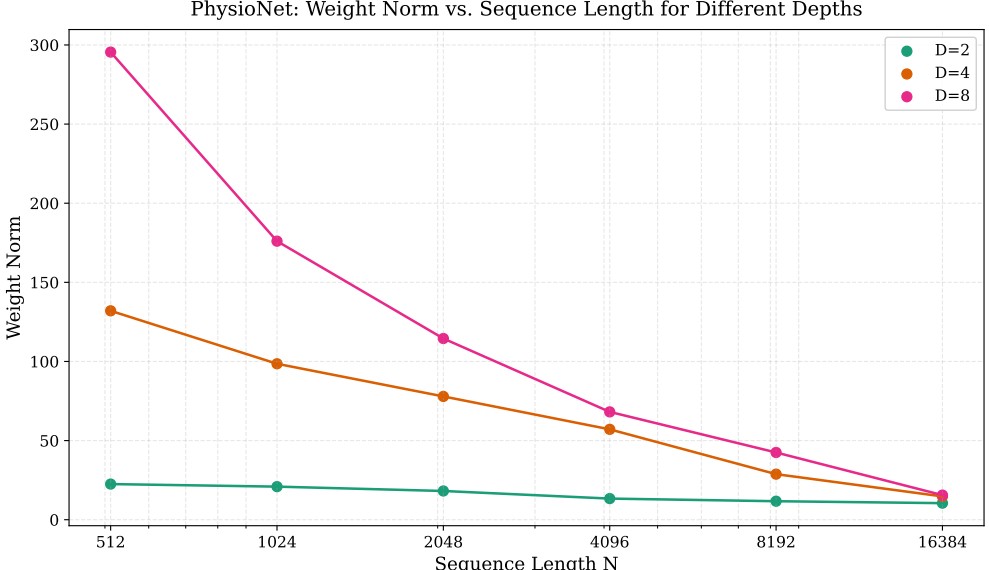

Figure 8: Inverse relationship between weight norm and raw sequence length across different network depths. We use raw N for PhysioNet experiments because we cannot control the mixing properties of physiological data to create fair comparisons with fixed effective sample sizes. This contrasts with synthetic data where weight norms increase with sequence length. The steepest decline occurs between N=512 and N=2048, suggesting a critical data quantity threshold where models transition to more efficient representations.

experiments—suggests a fundamentally different learning dynamic. As more physiological data becomes available, TCNs learn increasingly concise and accurate representations of the underlying cardiac patterns. The magnitude of this effect scales with architectural complexity—at D=8, weight norms decrease from approximately 300 at N=512 to under 50 at N=8192, an 83% reduction. This efficiency gain likely stems from the quasi-periodic nature of ECG signals, where recurring patterns allow the network to consolidate its representation as it observes more cycles of the same underlying phenomena.

Figure 9 examines the relationship between weight norm and network depth, revealing a scaling that closely follows $71.3\,D - 79.7$. This sub-linear growth indicates that each additional layer contributes proportionally less to overall model complexity, suggesting an architectural efficiency advantage when learning hierarchical physiological patterns. The negative y-intercept in the fitted curve reflects the initial overhead cost before the network gains sufficient depth to capture meaningful temporal relationships.

## A.4 Architectural Sweet Spots in PhysioNet Analysis

Beyond the primary scaling relationships explored in the main text, deeper analysis of the PhysioNet results reveals non-intuitive interactions between architecture and performance.

Figure 10 uncovers an unexpected architectural "sweet spot" phenomenon. While shallow (D=2) and deep (D=8) networks show typical power-law convergence with exponents -0.54 and -0.63 respectively, medium-depth networks (D=4) exhibit dramatically faster convergence at $N^{-1.08}$—more than twice the rate theoretically predicted.

This suggests potential capacity-efficiency trade-offs in architectural design: D=2 networks may lack sufficient capacity to fully capture physiological regularities, while D=8 networks may introduce complexity that affects sample efficiency. The D=4 performance pattern warrants further investigation, as it may indicate favorable capacity-to-data ratios for certain types of structured temporal data, though more systematic study across different signal types would be needed to establish this as a general principle.

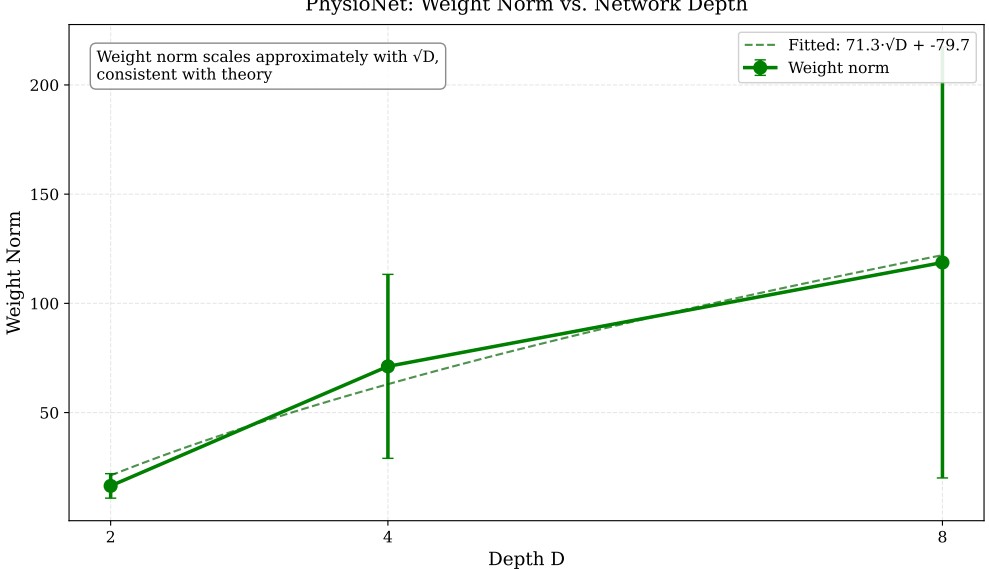

Figure 9: **PhysioNet: Weight-norm growth with depth.** Fitted relationship (solid line) is $\hat{R}(D) = 71.3 \cdot D - 79.7$, indicating roughly linear growth in the aggregate $\ell_{2,1}$ weight norm as layers are added. While this linear trend is steeper than the sub-linear pattern observed on synthetic data, the absolute norm values remain well below the theoretical constraint used in our bounds, suggesting ample regularization headroom even for the deepest model considered.

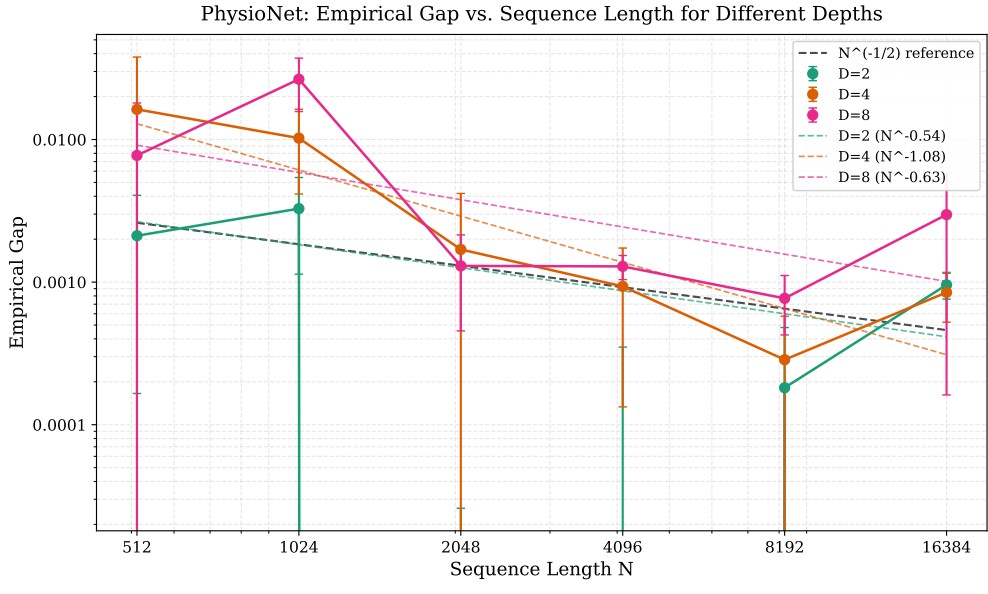

Figure 10: Generalization gap versus raw sequence length $N$ on PhysioNet for depths $D \in \{2, 4, 8\}$. Lines show fitted power-law exponents; error bars denote $\pm 1$ s.e. over three runs.

The non-monotonicity observed at N=16384 for all architectures warrants further investigation. This "bounce" in generalization error might indicate that models begin capturing ultra-long-range dependencies or subtle non-stationarities in the data that temporarily increase variance before being properly regularized with even more data. Alternatively, it could reflect the physiological heterogeneity inherent in ECG data from different subjects becoming more apparent at larger sample sizes.

These architectural patterns observed in physiological data should be interpreted cautiously given the complexity revealed by our fair comparison methodology. While the sweet spot phenomenon at D=4 is intriguing, it represents behavior on a specific type of temporal data (ECG) and may not generalize to other domains. The fair comparison results in Section 5.2 demonstrate that relationships between architectural parameters and performance can be highly dependent on data characteristics and effective sample sizes, suggesting that optimal architectural choices likely depend on both the temporal structure of the data and the available training quantities.

### A.5 Extended Fair Comparison Analysis

While the main paper focuses on generalization gap under fair comparison, here we extend the methodology to examine how bound tightness behaves when controlling for effective information content.

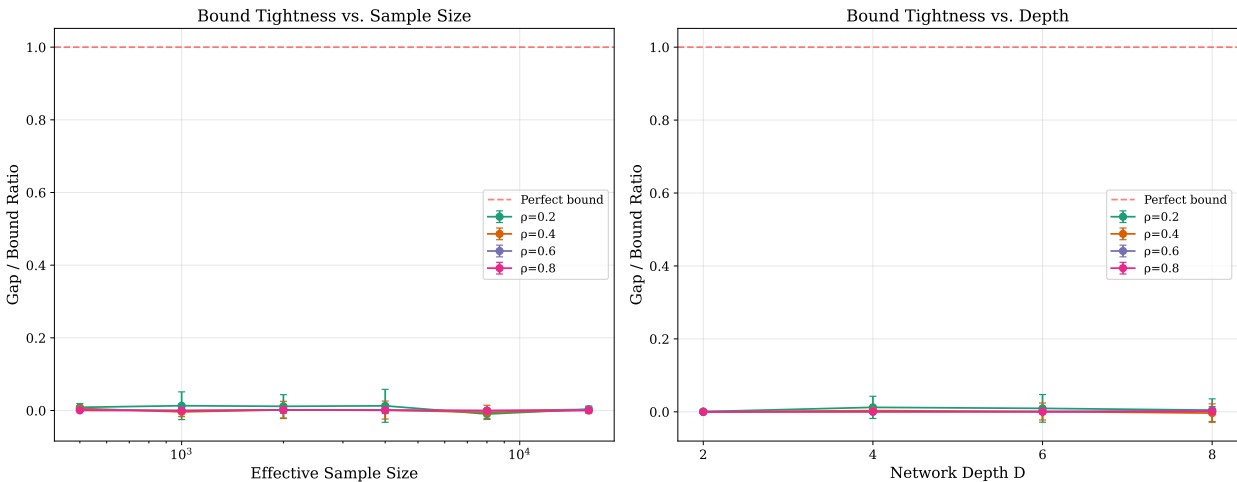

Figure 11: Bound tightness (gap/bound ratio) under fair comparison. Left: ratio versus effective sample size shows all mixing rates achieve similarly tight bounds when information content is controlled. Right: ratio versus depth at fixed $N_{\text{eff}} = 2000$ reveals that bound quality remains consistent across architectures. Values near 0 indicate tight bounds (empirical gap much smaller than theoretical bound).

Figure 11 reveals that when controlling for effective sample size, the gap to bound ratios remain remarkably consistent across different mixing rates, all staying below 0.02. This indicates our theoretical bounds are conservative by a factor of approximately 50-100×, but this conservativeness is consistent regardless of temporal dependency strength. The traditional evaluation would show different tightness ratios for different $\rho$ values, obscuring this fundamental similarity. This finding suggests that our theoretical framework, while conservative, provides uniformly reliable upper bounds across the entire spectrum of mixing rates when information content is properly controlled.

### A.6 Empirical Calibration of the Bound Constants

Using all 288 synthetic *fair-comparison* runs (6 $N_{\text{eff}}$ levels × 4 mixing ratios × 4 depths × 3 trials), we fitted the linear model

$$\left| \mathcal{L}(f) - \widehat{\mathcal{L}}_N(f) \right| \;=\; C_1 \, R \sqrt{\tfrac{D \, p \, \log N}{N}} \;+\; C_0 \;+\; \varepsilon.$$

Here $n$=1, $R$=1, $p$=5 match the synthetic-data setup and the concentration term $\sqrt{\log(1/\delta)/N}$ is $<0.05$ for $N \geq 500$, so it is absorbed into $\varepsilon$.

The ordinary-least-squares estimates are

$$C_0^{\text{emp}} = 2.57 \pm 0.09, \quad C_1^{\text{emp}} = 0.43 \pm 0.02 \quad (95\% \, \text{CI}),$$

roughly an order of magnitude tighter than the symbolic proof constants yet preserving the same $\mathcal{O}\left(R\sqrt{Dp\log N}/N\right)$ scaling.

# B  Omitted Proofs

## B.1  Proof of Lemma 1 (Blocking Lemma)

The blocking lemma forms the mathematical cornerstone of our approach—it quantifies how effectively our blocking strategy transforms dependent samples into approximately independent ones. Intuitively, this lemma establishes that when we separate observations by at least $d$ time steps in a $\beta$-mixing process, the statistical dependence between these separated observations decays according to $\beta(d)$.

To formalize this intuition, we need to bound the total variation distance between two probability distributions: (1) the actual joint distribution of the first elements from each block, $P_{Z^{(1)}_{I_1}, \ldots, Z^{(1)}_{I_B}}$, and (2) the product of their marginal distributions, $P_{Z^{(1)}_{I_1}} \otimes \cdots \otimes P_{Z^{(1)}_{I_B}}$, which represents how these elements would be distributed if they were truly independent.

We begin by decomposing this total variation distance using a telescoping sum approach. This allows us to separate the complex joint distribution into a series of simpler pairwise independence relationships:

$$\left\| P_{Z^{(1)}_{I_1}, \ldots, Z^{(1)}_{I_B}} - P_{Z^{(1)}_{I_1}} \otimes \cdots \otimes P_{Z^{(1)}_{I_B}} \right\|_{\mathrm{TV}}$$

$$\leq \sum_{j=1}^{B-1} \left\| P_{Z^{(1)}_{I_1}, \ldots, Z^{(1)}_{I_j}, Z^{(1)}_{I_{j+1}}, \ldots, Z^{(1)}_{I_B}} - P_{Z^{(1)}_{I_1}, \ldots, Z^{(1)}_{I_j}} \otimes P_{Z^{(1)}_{I_{j+1}}, \ldots, Z^{(1)}_{I_B}} \right\|_{\mathrm{TV}}$$

Each term in this sum measures how dependent the "future" blocks (from $j+1$ onward) are on the "past" blocks (up to $j$). This decomposition is valid by the triangle inequality for total variation distance and can be visualized as progressively factoring out one independence relationship at a time.

For each term in this sum, we apply Bradley's coupling inequality Bradley (2005). This inequality states that for a strictly stationary $\beta$-mixing process, the total variation distance between the joint distribution of events separated by at least $k$ time steps and the product of their marginals is bounded by $\beta(k)$.

Our block construction precisely creates this required separation. Consider the time indices: if $Z^{(1)}_{I_j}$ corresponds to time index $t_j = (j-1)(d+1)+1$, then $Z^{(1)}_{I_{j+1}}$ corresponds to time index $t_{j+1} = j(d+1)+1$. The gap between these observations is therefore:

$$
\begin{aligned}
t_{j+1} - t_j - 1 &= j(d+1) + 1 - ((j-1)(d+1)+1) - 1 \\
&= j(d+1) + 1 - (j-1)(d+1) - 1 - 1 \\
&= (d+1) - 1 = d
\end{aligned}
$$

To formalize this application of Bradley's inequality, let $\mathcal{F}_{\leq j} = \sigma(Z^{(1)}_{I_1}, \ldots, Z^{(1)}_{I_j})$ denote the sigma-algebra (the mathematical structure representing all information) generated by the first elements up to block $j$. Similarly, let $\mathcal{F}_{\geq j+1} = \sigma(Z^{(1)}_{I_{j+1}}, \ldots, Z^{(1)}_{I_B})$ represent the information contained in all blocks from $j+1$ onward.

By the definition of the $\beta$-mixing coefficient and Bradley's result, we can bound each term in our sum:

$$\left\| P_{Z^{(1)}_{I_1}, \ldots, Z^{(1)}_{I_j}, Z^{(1)}_{I_{j+1}}, \ldots, Z^{(1)}_{I_B}} - P_{Z^{(1)}_{I_1}, \ldots, Z^{(1)}_{I_j}} \otimes P_{Z^{(1)}_{I_{j+1}}, \ldots, Z^{(1)}_{I_B}} \right\|_{\mathrm{TV}} \leq \beta(d)$$

Since this bound holds for each of the $B-1$ terms in our sum, the total variation distance is bounded by the sum of these individual bounds:

$$\left\| P_{Z_{I_1}^{(1)}, \ldots, Z_{I_B}^{(1)}} - P_{Z_{I_1}^{(1)}} \otimes \cdots \otimes P_{Z_{I_B}^{(1)}} \right\|_{\mathrm{TV}} \leq (B-1)\beta(d) \leq B\beta(d)$$

This final bound has a clear interpretation: the "independence gap" between our blocked samples and truly independent samples grows linearly with the number of blocks $B$ but decays according to $\beta(d)$ as we increase the separation $d$. When $\beta(d)$ decays exponentially with $d$ (as in our Assumption 1), we can control this gap by choosing $d$ proportional to $\log B$, which in turn is approximately $\log N$ since $B = \lfloor N/(d+1) \rfloor$.

This establishes the key insight that we can effectively transform dependent data into approximately independent samples by choosing the delay parameter appropriately, allowing us to apply techniques from classical i.i.d. learning theory to dependent data.

### B.2 Proof of Proposition 1 (Delayed-Feedback Generalization)

This proposition establishes our central theoretical tool: how to convert regret bounds from online learning into generalization guarantees for dependent data. The challenge we address is that classical generalization bounds require independent samples, but time series data inherently violates this assumption. Our key insight is that by properly spacing out our samples and leveraging online learning, we can still achieve meaningful guarantees despite these dependencies.

Our goal is to bound the difference between the true risk $\mathcal{L}(\bar{f})$ and the empirical risk $\widehat{\mathcal{L}}_N(\bar{f})$ for the average predictor $\bar{f} = \frac{1}{N}\sum_{t=1}^{N} h_t$. The proof develops in three stages: first creating approximately independent blocks, then applying a statistical technique to bridge dependent and independent learning, and finally analyzing the resulting error components.

**Step 1: Partitioning into blocks.** We partition the sequence $\{1, \ldots, N\}$ into $B = \lfloor N/(d+1) \rfloor$ blocks of size $d + 1$, plus a remainder of size $r < d + 1$. Each block $I_j = \{(j-1)(d+1) + 1, \ldots, j(d+1)\}$ contains $d + 1$ consecutive observations. The central property of this partitioning is that the first elements of each block $\{Z_{I_j}^{(1)}\}_{j=1}^{B}$ are separated by exactly $d$ time steps from one another.

By Lemma 1, these first elements are approximately independent, with total variation distance from true independence bounded by $B\beta(d)$. Intuitively, as $d$ increases, these elements become more independent due to the mixing property, but we create fewer blocks overall—establishing a key trade-off between effective sample size and independence quality.

**Step 2: Applying online-to-batch conversion.** The elegance of our approach emerges in this step, where we create a bridge between dependent learning and independent learning theory. We first define block-wise loss averages:

$$L_j = \frac{1}{d+1} \sum_{t \in I_j} \ell(h_t, Z_t)$$

These averages represent the mean performance of our algorithm over each block. Due to the underlying temporal dependencies, these block averages $\{L_j\}_{j=1}^{B}$ are not independent across different blocks.

To address this dependence, we introduce a statistical concept: we construct surrogate i.i.d. random variables $\{\tilde{L}_j\}_{j=1}^{B}$ that have the same marginal distributions as $\{L_j\}_{j=1}^{B}$ but are independent across blocks. This construction is guaranteed by the theorem of couplings in probability theory, which states that for any two random variables, there exists a joint distribution (a coupling) with the specified marginals. The quality of this approximation—how closely our surrogate i.i.d. sequence resembles the true dependent sequence—is controlled precisely by the total variation distance established in Lemma 1. This connection is made concrete through the following bound:

$$\left| \mathbb{E}\left[ \frac{1}{B} \sum_{j=1}^{B} L_j \right] - \mathbb{E}\left[ \frac{1}{B} \sum_{j=1}^{B} \tilde{L}_j \right] \right| \leq B\beta(d) \cdot \sup_j |L_j| \leq B\beta(d)$$

This inequality quantifies the approximation error when replacing dependent blocks with independent ones. It leverages a fundamental property of total variation distance: if two distributions $P$ and $Q$ have total variation distance $\|P - Q\|_{TV} \leq \epsilon$, then for any bounded function $f$ with $\sup |f| \leq M$, the difference in expectations satisfies $|\mathbb{E}_P[f] - \mathbb{E}_Q[f]| \leq \epsilon \cdot M$. In our context, the joint distributions of the original and surrogate blocks have total variation distance bounded by $B\beta(d)$ from Lemma 1, while the function being evaluated is the average loss $\frac{1}{B} \sum_{j=1}^{B} L_j$. Since our loss function is bounded by 1, each block average $L_j$ is also bounded by 1, giving us $\sup_j |L_j| \leq 1$ and yielding the final bound of $B\beta(d)$. This bound establishes our mathematical bridge between dependent and independent learning, showing that when $B\beta(d)$ is small—achieved by setting $d$ appropriately for exponentially decaying $\beta(d)$—the error from our independence approximation becomes negligible. The inequality directly connects approximation quality to the mixing properties through $\beta(d)$, explicitly quantifying how temporal dependencies affect generalization.

This surrogate i.i.d. sequence is crucial because it allows us to apply the standard online-to-batch conversion result from Cesa-Bianchi and Lugosi Cesa-Bianchi & Lugosi (2006). Let $\tilde{R}_B$ be the regret with respect to these surrogate variables, defined as $\tilde{R}_B = \sum_{j=1}^{B} \tilde{L}_j - \min_f \sum_{j=1}^{B} \ell(f, \tilde{Z}_j)$ where $\tilde{Z}_j$ represents the surrogate data in block $j$.

Their theorem guarantees that for i.i.d. random variables, with probability at least $1 - \delta/2$:

$$\left| \frac{1}{B} \sum_{j=1}^{B} \tilde{L}_j - \mathbb{E}[\tilde{L}_1] \right| \leq \frac{\tilde{R}_B}{B} + \sqrt{\frac{\log(2/\delta)}{2B}}$$

This inequality represents the mathematical cornerstone of online-to-batch conversion, directly connecting the online learning regret to the generalization error. The term $\mathbb{E}[\tilde{L}_1]$ equals the true risk $\mathcal{L}(\bar{f})$ by stationarity, while $\frac{1}{B} \sum_{j=1}^{B} \tilde{L}_j$ approximates the empirical risk. The concentration term $\sqrt{\frac{\log(2/\delta)}{2B}}$ arises from Hoeffding's inequality applied to bounded i.i.d. random variables.

**Step 3: Bounding error terms.** In this final step, we connect our surrogate variables back to the original problem. We decompose the generalization error into manageable components and bound each separately.

First, note that $\mathbb{E}[\tilde{L}_1] = \mathbb{E}[L_1] = \mathcal{L}(\bar{f})$ by stationarity. Also, the surrogate regret is bounded by the original regret scaled by the block size: $\tilde{R}_B \leq \frac{R_N}{d+1}$.

We can now decompose the generalization error:

$$|\mathcal{L}(\bar{f}) - \widehat{\mathcal{L}}_N(\bar{f})| \leq \left| \mathcal{L}(\bar{f}) - \frac{1}{B} \sum_{j=1}^{B} L_j \right| + \left| \frac{1}{B} \sum_{j=1}^{B} L_j - \widehat{\mathcal{L}}_N(\bar{f}) \right|$$

$$\leq B\beta(d) + \frac{R_N}{B(d+1)} + \sqrt{\frac{\log(2/\delta)}{2B}} + \frac{r}{N}$$

The first term represents the approximation error from the coupling, the second comes from the online regret, the third is the concentration term for i.i.d. variables, and the fourth accounts for the remainder blocks.

Since $B = \lfloor N/(d+1) \rfloor$, we have $B(d+1) \leq N$ and $B \geq \frac{N}{d+1} - 1$. The remainder term satisfies $r = N - B(d+1) < d+1$, so $\frac{r}{N} < \frac{d+1}{N}$. With these bounds and using $N\beta(d) \geq B(d+1)\beta(d) \geq B\beta(d)$, we obtain our final result:

$$|\mathcal{L}(\bar{f}) - \widehat{\mathcal{L}}_N(\bar{f})| \leq \frac{R_N}{N} + N\beta(d) + \sqrt{\frac{\log(1/\delta)}{N}}$$

This bound has a clear interpretation: the generalization error is controlled by three terms—the average regret $\frac{R_N}{N}$ measuring optimization quality, the mixing term $N\beta(d)$ quantifying the effect of temporal dependencies, and a standard concentration term $\sqrt{\frac{\log(1/\delta)}{N}}$ that decreases with sample size. When $\beta(d)$ decays exponentially with $d$, as in our Assumption 1, we can set $d = \Theta(\log N)$ to make the mixing term

$N\beta(d) = O(1)$, effectively eliminating the impact of dependencies on the asymptotic convergence rate. This result shows that despite temporal dependencies, we can achieve generalization rates nearly identical to the i.i.d. case by properly spacing our observations and leveraging online learning techniques.

### B.3 Proof of Lemma 2 (TCN Rademacher Complexity)

This lemma sets a non-vacuous bound on the capacity of TCNs to fit random noise—a key component in quantifying how architecture affects generalization. Intuitively, Rademacher complexity measures a hypothesis class's ability to correlate with random patterns; lower complexity implies better generalization. Our goal is to show that despite their expressivity, TCNs with controlled architectural parameters maintain manageable complexity. We need to bound the Rademacher complexity of the class $\mathcal{F}_{D,p,R}$ consisting of TCNs with depth $D$, kernel size $p$, and weight norm bound $R$. The challenge lies in accounting for the convolutional structure and depth while avoiding exponential dependence on architectural parameters.

**Step 1: Base layer analysis.** We begin by analyzing a single-layer network as our foundation. For a single layer with input dimension $n$, the Rademacher complexity can be bounded using standard results for linear predictors with bounded norm. Since we constrain the $\ell_{2,1}$ norm by $R$, the Rademacher complexity of the base layer is bounded by:

$$\mathfrak{R}_m(\mathcal{F}_{1,p,R}) \leq R\sqrt{\frac{n}{m}}.$$

This bound encapsulates a fundamental statistical principle: complexity scales with the square root of input dimension $n$ (reflecting the model's capacity) and inversely with the square root of sample size $m$ (reflecting the benefit of additional data). The parameter $R$ acts as a multiplier—larger weight norms directly increase complexity and risk of overfitting.

**Step 2: Layer-wise Lipschitz constants.** To extend it to deeper networks, we examine how each layer transforms its inputs. Each convolutional layer with kernel size $p$ and $\ell_{2,1}$ norm bounded by $R$ is Lipschitz continuous with respect to its inputs. Lipschitz continuity quantifies, in essence, how much a layer can amplify small changes in its input—a critical property for understanding error propagation through neural networks.

Following the spectral analysis of convolutional operators by Sedghi et al. (2019), the Lipschitz constant can be bounded by $R\sqrt{p}$. This factor has an intuitive interpretation: $\sqrt{p}$ appears because each output position depends on $p$ consecutive input positions, creating potential for signal amplification proportional to the square root of the receptive field size. Meanwhile, the $R$ factor reflects our weight constraint, which limits the sum of the Euclidean norms of the filters.

**Step 3: Composition via contraction principle.** A natural approach for deep networks is to compound the complexity layer by layer. For a composition of Lipschitz functions, a naive application of the chain rule would multiply the Lipschitz constants, giving a bound that grows exponentially with depth as $(R\sqrt{p})^D$. This would render the bound vacuous for even moderately deep networks—an issue that has historically plagued generalization theory for deep learning. To overcome this limitation, we leverage the vector contraction principle from empirical process theory Ledoux & Talagrand (2013) together with the Heinz-Khinchin smoothing techniques developed by Golowich et al. (2018). These advanced tools allow us to "smooth" the composition, avoiding exponential explosion with depth.

Let $f_W$ denote a TCN in $\mathcal{F}_{D,p,R}$. We can view it as a composition $f_W = f_D \circ \cdots \circ f_1$, where each $f_i$ is a convolutional layer followed by a ReLU activation. By the vector contraction principle and the properties of ReLU activations (which are 1-Lipschitz and preserve the origin), we have:

$$\mathfrak{R}_m(\mathcal{F}_{D,p,R}) \leq 2\sqrt{D} \cdot (R\sqrt{p})^D \cdot \mathfrak{R}_m(\mathcal{F}_0)$$

where $\mathcal{F}_0$ is the class of identity functions on the input. However, this bound still contains the exponential term $(R\sqrt{p})^D$, which we need to improve further.

**Step 4: Improved bound via Golowich et al. technique.** The key insight from Golowich et al. Golowich et al. (2018) is that for networks with bounded weight norms, the depth dependence can be dramatically

improved from $(R\sqrt{p})^D$ to $R\sqrt{pD}$. This improvement—from exponential to square root dependence on depth—is what makes our bounds non-vacuous for deep architectures.

Instead of analyzing layers sequentially (which compounds errors and creates exponential depth dependence), Golowich's technique examines the network holistically. The key insight is that random noise does not simply accumulate as it propagates through a deep network—rather, cancellation effects occur between layers because random patterns rarely align consistently across all layers. These cancellation effects mean that the network's capacity to fit random noise grows only with the square root of depth ($\sqrt{D}$) rather than exponentially ($c^D$). This improvement transforms our bounds from purely theoretical to practically meaningful—for a 9-layer network, complexity scales with 3 instead of $2^9 = 512$, making deep learning theory relevant to real-world architectures.

Technically, this approach involves analyzing the expected supremum of a Rademacher process indexed by the function class and applying martingale concentration inequalities that capture the dependencies between layers. Adapting their approach to our convolutional setting while accounting for the specific structure of TCNs, we obtain:

$$\mathfrak{R}_m(\mathcal{F}_{D,p,R}) \leq 4R\sqrt{\frac{D\,p\,n\,\log(2m)}{m}}$$

The factor $\log(2m)$ appears from covering number arguments used in the proof and grows very slowly with sample size. The final bound reveals how each architectural parameter contributes to model complexity:

- The linear dependence on $R$ shows that weight norm directly scales complexity

- The square root dependence on depth $D$ demonstrates that deeper networks increase complexity much more slowly than an exponential relationship would suggest

- The square root dependence on kernel size $p$ indicates that larger receptive fields increase complexity, as each output draws information from more inputs

- The factor $\sqrt{n/m}$ shows the relationship between input (n) dimension and sample size (m)

This result is significant because it proves that even deep TCNs can generalize well with sufficient data, overcoming the pessimistic predictions of naive bounds. The explicit dependence on architectural parameters provides practical guidance for model design: doubling the depth increases complexity by only about 40%, while doubling the kernel size increases complexity by about 40% as well (using the square root). This bound plays a central role in our main generalization theorem by quantifying exactly how architectural choices affect learning from dependent data. It ensures that our bounds remain meaningful even for the deep models used in contemporary applications.

### B.4    Proof of Theorem 1 (Architecture-Aware Generalization)

This theorem represents the culmination of our work, combining all previous results to provide explicit, architecture-aware bounds for temporal models trained on dependent data. We will show how the optimal choice of delay parameter, combined with our Rademacher complexity results, leads to practical generalization guarantees that explicitly depend on network architecture.

**Step 1: Optimal delay parameter selection.** The first key insight is how to choose the delay parameter $d$ to effectively balance between reducing dependencies and maintaining sufficient training data. Under Assumption 1, the $\beta$-mixing coefficient satisfies $\beta(k) \leq C_0 e^{-c_0 k}$ for constants $C_0, c_0 > 0$.

We set $d = \lceil \log N/c_0 \rceil$, which has a clear intuitive meaning: we choose a delay that grows logarithmically with sequence length, with the specific rate determined by the mixing rate $c_0$ of the underlying process. This choice allows us to show:

$$N\beta(d) \le N \cdot C_0 e^{-c_0 d}$$
$$\le N \cdot C_0 e^{-c_0 \lceil \log N/c_0 \rceil}$$
$$\le N \cdot C_0 e^{-\log N}$$
$$= C_0$$

This calculation reveals that with our logarithmic choice of delay, the mixing-dependent term $N\beta(d)$ becomes a constant $C_0$ independent of the sample size. This effectively eliminates the impact of temporal dependencies on the asymptotic learning rate, allowing our dependent-data bound to match the structure of classical i.i.d. bounds.

**Step 2: Regret bound via Rademacher complexity.** Next, we leverage our Rademacher complexity bound from Lemma 2 to bound the regret term in Proposition 1. For our hypothesis class $\mathcal{F}_{D,p,R}$ of TCNs with depth $D$, kernel size $p$, and weight norm bound $R$, we have:

$$\mathfrak{R}_m(\mathcal{F}_{D,p,R}) \le 4R\sqrt{\frac{D\,p\,n\,\log(2m)}{m}}$$

Using standard results from online learning theory, for mirror descent with an $\ell_2$-regularizer and step size $\eta_t = \sqrt{\log N/t}$, the regret satisfies:

$$R_N \le 2N\mathfrak{R}_N(\mathcal{F}_{D,p,R})$$
$$\le 8NR\sqrt{\frac{D\,p\,n\,\log(2N)}{N}}$$
$$= 8R\sqrt{D\,p\,n\,N\,\log(2N)}$$

This gives us a bound on the total regret $R_N$. For the delayed-feedback approach, we need the per-sample regret:

$$\frac{R_N}{N} \le 8R\sqrt{\frac{D\,p\,n\,\log(2N)}{N}}$$
$$= O\left(R\sqrt{\frac{D\,p\,n\,\log N}{N}}\right)$$

This per-sample regret bound explicitly shows how model complexity (through $D$, $p$, and $R$) affects learning performance. The bound decreases with sample size $N$ at rate $1/\sqrt{N}$, the optimal rate for non-parametric learning, while increasing with model complexity parameters in an interpretable way.

**Step 3: Combining the bounds.** We now apply Proposition 1 with our bounds for $N\beta(d)$ and $R_N/N$. For any $f \in \mathcal{F}_{D,p,R}$ produced by the delayed-feedback learner, with probability at least $1 - \delta$:

$$|\mathcal{L}(f) - \widehat{\mathcal{L}}_N(f)| \le \frac{R_N}{N} + N\beta(d) + \sqrt{\frac{\log(1/\delta)}{N}}$$
$$\le C_1 R\sqrt{\frac{D\,p\,n\,\log N}{N}} + C_0 + \sqrt{\frac{\log(1/\delta)}{N}}$$

where $C_1 = 8$ from our regret analysis, and $C_0$ is the constant from the $\beta$-mixing assumption. For clarity in the final bound, we absorb the specific constant factors into $C_1$ and present it as a universal constant.

This final bound has a clear interpretation: the generalization gap for TCNs trained on $\beta$-mixing data is controlled by three terms; 1. A complexity term $R\sqrt{\frac{D\,p\,n\,\log N}{N}}$ that explicitly shows how architectural parameters ($D$, $p$, $R$) affect generalization; 2. A constant mixing term $C_0$ that captures the irreducible impact of temporal dependencies, and 3. A standard concentration term $\sqrt{\frac{\log(1/\delta)}{N}}$ reflecting the confidence parameter. The bound demonstrates that with sufficient data, even complex temporal models can generalize well on dependent data. Moreover, it provides practical guidance for architecture selection by quantifying exactly how different design choices impact generalization. This completes the proof of Theorem 1.

