# OpenReview forum: "Architecture-Aware Generalization Bounds for Temporal Networks: Theory and Fair Comparison Methodology"
_TMLR — Rejected by TMLR_

### Review · Reviewer_152z · 2025-09-10

**Summary Of Contributions:**

On the theory front, the submission studies the generalization performance of TCNs [temporal convolution networks] for stationary data that is $\beta$-mixing (i.e., for which dependies decrease exponentially fast in some way over time). The main theoretical result (Theorem 1) provides a generalization bound composed of three terms, a regret term of order $\sqrt{D/N}$, where $N$ is the sample size and $D$ the depth of the TCN, a constant term $C_0$ due to data not being i.i.d. but $\beta$-mixing, and a mild $1/\sqrt{N}$ due to deviation inequalities.

On the methodological front, the submission recommends studying generalization errors based on so-called effective sample sizes rather than the raw sample sizes $N$.

**Additional Comments:**

I list here smaller-order comments, along the text, in no particular order:
- Abstract: $N_{\mbox{eff}}$ is not defined at this stage
- The way citations are incorporated in the text must be carefully reviewed throughout (sentences must be readable even with citations, and there should be no duplicates of authors' names)
- End of page 2: The $W$-norms sentences are difficult to understand at this stage of the submission
- Right before Section 3.1: Consider mentioned already here (and even in the abstract or earlier in the submission) that data is stationary
- Section 3.3: Hazan (2016) is a late reference for regret; rather mention a monograph or a much earlier article
- Page 7: Syntax/structure of the sentence ``When $d$ is chosen ... negligibly small''
- Proposition 1: I would have expected the performance of $\bar{f}$ to be compared to some best $f$ in some class
- Pages 3 and 9: CNNs--explain the acronym (just as was done for TCNs)
- Page 9: A picture would help to understand why TCNs are great because ``deeper layers to capture longer-range dependencies
without a proportional increase in parameters''
- Page 11: Define formally AR(1) processes with a lag-1 autocorrelation $\rho$; recall formally what an efficient sample size is
- End of page 11 - beginning of page 12: This methodology is an interesting remark but I see it rather as a remark given that theoretical bounds are not in terms of $N_{\mbox{eff}}$
- Sectin 5.2: Give more indications on the setting: exact specification of the AR(1) process, how $d$ was picked, etc.
- Sections 6 and 7: They contain repetitive material; consider reading again the entire text of the submission and making it more compact by erasing duplicate comments

**Audience:**

Yes

**Audience Explanation:**

Some claims, like that doubling the depth of a TCN only requires 4 times more data to get the generalization error, even in non-i.i.d. scenarios where data is stationary and $\beta$-mixing, is an interesting claim for the TMLR audience, as it would guide the construction of TCNs.

(My only issue is that, as discussed above, I cannot see how this claim comes from Theorem 1.)

**Claims And Evidence:**

No

**Claims Explanation:**

The proof sketches, the intuitions conveyed, and various remarks made in the submission make it enjoyable to read.

Alas, *first*, some parts of the submission lack formalization and I would need additional details to be able to make a fully informed recommendation (in either direction: I have no opinion for now, I would rather like to understand the results and their proofs).

The algorithm and the proof schemes:
- The text is unclear on what whose generalization error is actually studied. Theorem 1 is about any function f in a class, while one of its supporting results, Proposition 1, is about some average of hypotheses picked by some auxiliary adversarial-learning algorithm (that the learner picks?). Also, there is a blocking scheme, but I believe it's only used for the analysis. Is it the same for the adversarial-learning strategy? It may be that the parameter $d$ is only used for the analysis and not for defining the strategy, in which case taking it as a function of the unknown $c_0$ rate would not be an issue. In that case, consider not mentioning $d$ in the statement of the theorem. Otherwise, I think there would be a severe adaptivity issue: the generalization bound would only hold knowing the rate $c_0$ (and there's no reason for knowing it).
- I tried to read the proofs (located in pages 27--33) to get a better understanding of what is studied. Alas, these proofs come typically more with intuitions or undetailed claims (both similar to text in the main body) and lack formalization, though their results are believable. I provide specific examples below.

About the proofs:
- As indicated, they contain repetitive material and I would have expected some rather dry mathematical text in Appendix B
- Proof of Lemma 1: it would be useful to re-state Bradley's coupling inequality to make the text self-contained; the (trivial) calculation of $t_{j+1}-t_j-1$ has no reasons to be detailed; on the contrary, the application of Bradley's inequality to get the $\beta(d)$ bound must be detailed
- Proof of Proposition 1: Lemma 1 is used in Step 2, but Lemma 1 only involves the first steps of each block, while the random variables at stake here depend on all data of the block---I do not see how Lemma 1 can be applied; I cannot relate the left-hand side and the right-hand side of the last display of Step 2: the left-hand side compares an average of B i.i.d. random variables to their mean, which is directly smaller than something of order $\sqrt{\ln(1/\delta)/B}$ by Hoeffding's inequality, I don't see where the regret term comes from in the bound; finally, it seems critical to me to detail why the term $\mathbb{E}[\tilde{L}_1]$ equals the true risk $\mathcal{L}(\bar{f})$
- Proof of Lemma 2: I'm not a specialist of neural networks, so I cannot evaluate well this proof; that being said, it starts with comments and intuition and what seems the main technical point (``Adapting their approach to our convolutional setting while accounting for the specific structure of TCNs, we obtain: [...]'') is not detailed, while it should be in my eyes
- Proof of Theorem 1: Similar comments apply; the inequality $N \, \beta(d) \leq C_0$, already proved in the main body, is detailed again, while the critical part is not (``Using standard results from online learning theory, for mirror descent with an $\ell_2$-regularizer and step size $\eta_t = \sqrt{\ln N/t}$, the regret satisfies [...]'')---I do not see how $R_N$ is bounded by $2N$ times some Rademacher complexity

*Second*, I wonder about the implications of Theorem 1. The text (see top of page 11) discusses the first term in the generalization bound, which vanishes with $N$, but the second term, the constant $C_0$, does not vanish and could be large (even larger than the left-hand side, which is bounded). I see no discussion about this in the text, while to me, the existence of $C_0$ could lead to a void bound in Theorem 1.

**Requested Changes:**

See the comments above; I would like the exposition of Section 4 to be improved (to separate the target quantities from the auxiliary results), the implications of the constant $C_0$ in Theorem 1 to be discusssed, and all proofs of Appendix B to be made more formal (and to not contain comments or intuitions that are repetitive with text in the main body)

---

> ### Author Response · Authors · 2025-09-17
>
> Thank you for your thorough feedback (5000 chars)
>
> **(1) $C_0$ constant and $\beta$-mixing**
> You correctly note $C_0$ depends on mixing properties. With optimal delay $d^* = \lceil\log N/c_0\rceil$:
> $$N\beta(d^*) \leq N \cdot C_0 e^{-c_0\lceil\log N/c_0\rceil} \leq N \cdot C_0 e^{-\log N} = C_0$$
> This makes $C_0$ sample size independent, eliminating the $1/\sqrt{N}$ term.
>
> **(2) Function scope in Theorem 1 (ambiguity)**
> Thrm 1 applies specifically to $\bar{f} = \frac{1}{N}\sum_{t=1}^{N} h_t$ output by the delayed-feedback learning algorithm, not arbitrary $f \in \mathcal{F}_{D,p,R}$. The bound holds only for this averaged hypothesis from the specific learning procedure, with algorithm dependence entering through regret $R_N$ capturing the online trajectory.
>
> Revisions: Specify $\bar{f}$ in Theorem 1; clarify Section 3.3 applies to averaged hypotheses; ; note proof analyzes specific averaged hypothesis.
>
> You raise a valid "severe adaptivity issue'', our bound requires an unknown $c_0$, which we did not adequately address as a limitation. We acknowledge this gap, for any conservative lower bound $\underline{c} \leq c_0$, setting $d = \lceil\log N/\underline{c}\rceil$ ensures:
> $$N\beta(d) \leq C_0 N^{1-c_0/\underline{c}} \leq C_0$$
> when $\underline{c} \leq c_0$, maintaining the $O(1)$ mixing term. $\underline{c}$ can be estimated from domain knowledge (ECG correlation half-lives $\rightarrow c_0 \approx 0.2-0.4$), empirical autocorrelation decay using $\underline{c} \approx -\log|\hat{\rho}(k)|$ with $\beta(k) \lesssim \rho^2(k)$, etc. This provides principled guidance validity, we'll clarify this in the revision.
>
> **(3) Technical Proofs - Lemma 1 (Blocking)**
> **Block separation:** First element of block $j$ at index $(j-1)(d+1) + 1$; block $j+1$ at $j(d+1) + 1$. Separation: $[j(d+1) + 1] - [(j-1)(d+1) + 1] - 1 = d$.
>
> **Bradley's Inequality:** For $\beta$-mixing processes, if
> $A \in \sigma(Z_1,\ldots,Z_s)$ and $B \in \sigma(Z_{s+d+1}, Z_{s+d+2}, \ldots)$,
> then $|\Pr(A \cap B) - \Pr(A)\Pr(B)| \leq \beta(d)$.
> Our telescoping decomposition measures dependence between past blocks
> $\{1,\ldots,j\}$ and future blocks $\{j+1,\ldots,B\}$.
> Since first elements are $d$-separated, each term is bounded by $\beta(d)$,
> yielding $B\beta(d)$.
>
> **(4) Proposition 1**
>
> **Coupling:** While $L_j = \frac{1}{d+1}\sum_{t \in I_j} \ell(h_t, Z_t)$ depends on all elements in block $j$, dependence between blocks is controlled by dependence between their first elements since blocks are separated by $d$ steps. We construct surrogate i.i.d. variables $\{\tilde{L}_j\}_{j=1}^B$ with same marginal distribution $\mu$ as $\{L_j\}$ by stationarity. The coupling lemma ensures total variation distance between joint distribution of $\{L_j\}$ and product distribution of $\{\tilde{L}_j\}$ is bounded by $B\beta(d)$ from Lemma 1.
>
> **Regret term:** originate through online-to-batch conversion (Cesa-Bianchi & Lugosi,2006). For surrogate i.i.d. variables:
> $$\left|\frac{1}{B}\sum_{j=1}^B \tilde{L}_j - E[\tilde{L}_1]\right| \leq \frac{\tilde{R}_B}{B} + \sqrt{\frac{\log(2/\delta)}{2B}}$$
> where $\tilde{R}_B \leq \frac{R_N}{d+1}$ since each block contains $d+1$ observations. The coupling error $\left|E[\frac{1}{B}\sum L_j] - E[\frac{1}{B}\sum \tilde{L}_j]\right| \leq B\beta(d)$ connects to the original sequence.
>
> **Why $E[\tilde{L}_1] = L(\bar{f})$:** By construction, $\tilde{L}_1$ has same marginal as $L_1 = \frac{1}{d+1}\sum_{t \in I_1} \ell(h_t, Z_t)$. By stationarity:
> $$E[L_1] = \frac{1}{d+1}\sum_{t \in I_1} E[\ell(h_t, Z_t)] = E[\ell(h_1, Z_1)] = L(\bar{f})$$
> Since $E[\tilde{L}_1] = E[L_1]$, we get $E[\tilde{L}_1] = L(\bar{f})$.
>
> **(5) Lemma 2 \& Theorem 1**
> **For Lemma 2:** We adapt Golowich et al. (2018) Thrm 1, which bounds Rademacher complexity for depth-$D$ networks under spectral norm constraints. Each TCN layer $\ell$ has weight tensor $W^{(\ell)} \in \mathbb{R}^{p \times r_{\ell-1} \times r_\ell}$ with $\|W^{(\ell)}\|_{2,1} \leq R$. The convolutional structure creates Lipschitz constant $R\sqrt{p}$ per layer (the $\sqrt{p}$ factor comes from kernel spanning $p$ steps). Applying depth-composition result:
> $$\mathfrak{R}_m(\mathcal{F}_{D,p,R}) \leq 4R\sqrt{\frac{Dpn\log(2m)}{m}}$$
> Their $\sqrt{D}$ scaling (not exponential) applies to causal convolutions because ReLU activations preserve parameter-sharing.
>
> **For Theorem 1 (Online Learning):** The regret bound $R_N \leq 2N \cdot \mathfrak{R}_N(\mathcal{F}_{D,p,R})$ follows from mirror descent analysis (Shalev-Shwartz,2012). With step size $\eta_t = \sqrt{\log N/t}$ and $\ell_2$-regularization, the factor $2N$ emerges from online-to-batch conversion relating cumulative regret to expected Rademacher complexity via Rademacher-Khinchine symmetrization.
>
> **Revisions:** 1) Revise Theorem 1 to specify $\bar{f}$ from delayed-feedback algorithm; 2) Add bound discussion with $c_0$ estimation; 3) Expand Appx B with Bradley derivations &coupling; 4) Parameter mappings from Golowich to TCN; 5) Restructure Sec 4

---

> ### Comment · Reviewer_152z · 2025-10-10
> **No rebuttal?**
>
> Dear authors, I cannot see the rebuttal (if any) to my report. You perhaps misconfigured the Readers field?

---

> > ### Comment · Action_Editor_Vi8v · 2025-10-10
> > **Please change visibility of rebuttal**
> >
> > Dear Authors,
> >
> > Reviewer 152z is right, the visibility of your rebuttal is hidden to the reviewers. Could you please share again with the right visibility?
> >
> > Thank you very much in advance

---

> > ### Author Response · Authors · 2025-10-10
> >
> > Can you check now? somehow it wasn't viewed for everyone
> >
> > we are sorry for that.

---

> > > ### Comment · Reviewer_152z · 2025-10-10
> > > **Rebuttal difficult to read**
> > >
> > > I can now see the rebuttal. However, I have to tell that it is difficult to read, given its telegraphic writing style and given many LaTeX formulas did not compile (while the rebuttals intended to the other two reviewers seem clearer). Also, there are statements that you corrected or retracted in these rebuttals, like the claim that the term $\mathbb{E}[\tilde{L}_1]$ is equal to the true risk $\mathcal{L}(\bar{f})$. Finally, it would have been helpful to adress my concerns in the same order, and address them all (at first sight, I cannot find an answer to all my concerns).
> > >
> > > For now, I therefore cannot assess this rebuttal.

---

> > > > ### Author Response · Authors · 2025-10-11
> > > >
> > > > We thank you for the feedback, we re-write our review, while also considering cYd2 review. We note that Reviewer cYd2 has identified fundamental errors in our original submission that we have acknowledged and corrected. Our response below addresses your concerns in light of these corrections.
> > > >
> > > > **Summary of Corrections (Addressing Both Reviews)**
> > > >
> > > > Following Reviewer cYd2's critique, we acknowledge four major errors: (1) Lemma 2 incorrectly dropped the product-of-layer-norms $R = \prod_{\ell=1}^D M(\ell)$; (2) the online-to-batch conversion requires convex, Lipschitz losses for Jensen's inequality; (3) concentration is $O(\sqrt{\log N/N})$ based on blocks $B$, not $O(1/\sqrt{N})$; (4) the step $R_N \le 2N\mathfrak{R}_N(\mathcal{F})$ was unjustified. The corrected bound becomes:
> > > >
> > > > $$\left|L(\bar{f}) - \hat{L}_N(\bar{f})\right| \lesssim \frac{R_N}{N} + R\sqrt{\frac{D\log N}{N}} + \sqrt{\frac{\log(1/\delta)\log N}{N}} + o(1)$$
> > > >
> > > > where $R = \prod_{\ell=1}^D M(\ell)$, under convex Lipschitz losses. We will revise the entire manuscript to reflect these corrections.
> > > >
> > > > **(1) Constant $C_0$ and $\beta$-mixing Dependence.**
> > > >
> > > > You correctly identify that $C_0$ depends on mixing properties. In the corrected framework, the dependence term arises from the block-level concentration. With $B = \lfloor N/(d+1)\rfloor$ blocks and optimal delay $d^* = \lceil \log N / c_0 \rceil$, the concentration scales as $O(1/\sqrt{B})$, which becomes $O(\sqrt{\log N/N})$ under exponential mixing with this choice of d^*.
> > > > The mixing coefficient enters through the choice of delay: stronger mixing (larger $c_0$) allows smaller $d^*$, yielding more blocks and tighter concentration.
> > > >
> > > > For practical application, $c_0$ can be estimated from empirical autocorrelation decay using standard time series methods. For AR(1) processes with parameter $\rho$, we have $c_0 = -\log(\rho)$. For physiological signals, domain knowledge (e.g., ECG correlation half-lives) provides reasonable estimates. While this introduces estimation uncertainty, the bound remains valid for any conservative choice: using $c \le c_0$ in $d = \lceil \log N / c \rceil$ ensures the mixing assumption holds, though possibly with a less efficient block count. We will clarify this estimation procedure and its implications in the revision.
> > > >
> > > > **(2) Function Scope in Theorem 1**
> > > >
> > > > You raise an important ambiguity that connects to the errors identified by Reviewer cYd2. Our Theorem 1 applies specifically to $\bar{f} = \frac{1}{N}\sum_{t=1}^N h_t$, the averaged hypothesis from delayed-feedback learning, not arbitrary $f \in \mathcal{F}_{D,p,R}$. The bound holds only for this averaged hypothesis, with algorithm dependence entering through regret $R_N$. However, as Reviewer cYd2 noted, we cannot justify the direct link $R_N \le 2N\mathfrak{R}_N(\mathcal{F})$ without additional assumptions. We will revise Theorem 1 to present it as algorithm-agnostic in $R_N$, which can be instantiated via standard OCO bounds. We will clarify: "Theorem 1 applies to averaged hypotheses from delayed-feedback learning; the bound analyzes the specific output of our learning algorithm, not arbitrary functions in the hypothesis class."
> > > >
> > > > **(3) Technical Proofs - Formal Revisions**
> > > >
> > > > We accept both reviewers' criticisms about proof rigor. We will revise Appendix B to remove intuitive commentary and provide formal derivations:
> > > >
> > > > *Lemma 1 (Blocking):* We will add explicit Bradley's coupling inequality and show the block separation calculation $\ell_{j-1} - \ell_j - 1$ step-by-step.
> > > >
> > > > *Proposition 1:* Following Reviewer cYd2's correction, we will replace the equality $E[L_j] = L(\bar{f})$ with Jensen's inequality, which requires assuming convex, Lipschitz losses. We will detail why block averages $\bar{L}_j$ have controlled variance and derive the concentration bound using the corrected $O(\sqrt{\log N/N})$ rate based on the number of blocks $B = \lfloor N/(d+1)\rfloor$.
> > > >
> > > > *Lemma 2 and Theorem 1:* Following Reviewer cYd2's correction, we will explicitly include $R = \prod_{\ell=1}^D M(\ell)$ and clarify that we achieve $\sqrt{D}$ scaling **alongside** this product term (as in Golowich et al., not beyond it). We will show how Golowich et al.'s result applies to TCNs with proper parameter mappings and norm constraints.

---

> > > > > ### Author Response · Authors · 2025-10-11
> > > > >
> > > > > **(4) Lemma 2 and Golowich Application.**
> > > > >
> > > > > As Reviewer cYd2 identified, we incorrectly claimed to eliminate the product-of-norms. The corrected Lemma 2 retains $R = \prod_{\ell=1}^D M(\ell)$ and achieves $\sqrt{D}$ depth dependence (improving from exponential, not eliminating $R$). However, our contribution extends beyond simply applying Golowich et al.'s result. We provide the first architecture-aware generalization analysis for **dependent sequences** by developing a complete framework combining five technical components: (1) a delayed-feedback blocking scheme that partitions sequences into blocks separated by delay $d$ while using all $N$ observations for training, (2) a coupling argument via Bradley's inequality showing that first elements of blocks are approximately independent with total variation distance bounded by $B\beta(d)$, (3) block-level concentration analysis yielding $O(\sqrt{\log N/N})$ rates, (4) Golowich et al.'s architecture-aware class complexity bound providing the $R\sqrt{D}$ term, and (5) online-to-batch conversion connecting regret to generalization under dependence.
> > > > >
> > > > > This framework is novel. Golowich et al. analyzed i.i.d. samples; we extend their architectural insights to $\beta$-mixing time series through the blocking-coupling-concentration pipeline. Our methodological contribution: the fair-comparison protocol using effective sample size $N_{\text{eff}} = N \cdot (1-\rho)/(1+\rho)$ for AR(1) processes provides a principled way to evaluate models under matched information content, revealing that temporal structure can enhance (not just hinder) generalization. We will restructure Section 4 to clearly delineate: (a) what we import from Golowich, (b) our theoretical contributions (blocking, coupling, concentration under mixing), and (c) our methodological contribution (fair-comparison evaluation).
> > > > >
> > > > > **(5) Regret Bound and Online Learning**
> > > > >
> > > > > As Reviewer cYd2 noted, we cannot justify $R_N \le 2N\mathfrak{R}_N(\mathcal{F})$ without additional assumptions. We will remove this claim and present Proposition 1 as algorithm-agnostic in $R_N$. When needed, we can instantiate $R_N$ using standard OCO bounds for specific algorithms (e.g., mirror descent with $\ell_2$-regularizer), but this is not claimed as a general result. We acknowledge this limitation in the revised manuscript.
> > > > >
> > > > > **Addressing the "Doubling Depth Requires 4x Data" Claim.**
> > > > >
> > > > > It follows from the corrected bound term $R\sqrt{D/N}$: maintaining constant error when depth increases from $D$ to $4D$ requires $N'' = 4N$ (assuming $R$ is controlled). However, this is **conservative theoretical guidance** from worst-case analysis. As empirical results show (Figure 3), actual requirements are lower when architectures exploit temporal structure. We will revise to: "Our theoretical analysis suggests doubling depth twice (from $D$ to $4D$) requires approximately 4x more data to maintain worst-case guarantees, though architectures well-matched to temporal structure may require less." We acknowledge this derives from the bound's scaling under the corrected framework, not a tight empirical prediction.
> > > > >
> > > > > **Requested Revisions Summary.**
> > > > >
> > > > > We will: (1) Correct Lemma 2 to include $R = \prod_{\ell} M(\ell)$ explicitly, clarifying we achieve $\sqrt{D}$ scaling alongside (not eliminating) this product term; (2) Add convexity and Lipschitz requirements to Proposition 1 with Jensen's inequality replacing the incorrect equality; (3) Correct concentration to $O(\sqrt{\log N/N})$ based on block count $B$; (4) Remove unjustified regret-Rademacher link and present Proposition 1 as algorithm-agnostic in $R_N$; (5) Clarify Theorem 1 scope: it applies to averaged hypotheses from our delayed-feedback algorithm, not arbitrary functions; (6) Add explicit $c_0$ estimation guidance and bounds under conservative choices; (7) Make all Appendix B proofs completely formal; (8) Add detailed Bradley's coupling inequality derivation, block separation calculations, and coupling argument showing how our blocking scheme creates approximately independent samples; (9) Expand our novel blocking-coupling-concentration pipeline showing how we extend architecture-aware analysis from i.i.d. to $\beta$-mixing sequences; (10) Add parameter mappings from Golowich's spectral/Frobenius norm framework to TCN filter norms; (11) Restructure Section 4 to clearly distinguish: (a) Golowich's contribution (architecture-aware class bound for i.i.d. data), (b) our theoretical contributions (blocking scheme, coupling via Bradley's inequality, block-level concentration under mixing, online-to-batch for dependent sequences), and (c) our methodological contribution (fair-comparison protocol via effective sample size). These revisions address all reviewers' concerns while clarifying that we develop a complete framework for dependent-sequence generalization that imports but substantially extends beyond Golowich's i.i.d. result.

---

> ### Comment · Reviewer_152z · 2025-10-12
> **Thank you for the rebuttal, which I could follow**
>
> I thank the authors for rewriting the rebuttal, which is now crystal clear. I also thank them for their honesty: they acknowledge all issues raised, and for most of them, they have (partial) solutions or a path to solutions.
>
> That being said, the corrections that will need to be performed are
> 1. heavy (basically, many statements and all proofs need to be rewritten so that reviewers would need to read again the submission to check again for correctness) and
> 2. substantial in the sense that the final claims will be significantly weaker than the original claims.
>
> All in all, I guess that it would be fair that this revised version be considered a new submission. This opinion is in line with the one by Reviewer cYd2.

---

### Review · Reviewer_zMST · 2025-10-03

**Summary Of Contributions:**

The paper derived an upper bound for the generalisation error of Temporal Convolutional Networks, on input sequence data that is temporally correlated and exhibits exponential \beta-mixing property. The dependence on depth is \sqrt{D}, making the bound non-vacuous for deeper architectures. For empirical analysis, the paper proposed a “fair comparison method” with an effective sample size measure, discounting the the raw sample size to reflect the reduced information contents of correlated sequences. Using the effective sample size suggests that temporal correlations may be beneficial to learning.

Strengths:

1.	The theory is clean, combining a series of techniques that leads to the \sqrt{D} dependency. I am not an expert in the area but I enjoyed reading the section and learned a number of things from the narrative.

2.	The effective sample size is an insightful new way to think about the performance of sequence models.


Weaknesses:

In general, I think stronger/more clearly presented empirical evidence is required to device recommendations in the style of “doubling model depth requires 4x more training data”.  For example, the \sqrt{D} dependence is not clearly obeyed by empirical results (Figure 3).

**Audience:**

Yes

**Audience Explanation:**

The generalisation bound is interesting for the theory community; and the idea of effective sample size could help us better understand sequence models.

**Claims And Evidence:**

No

**Claims Explanation:**

In general, I think stronger/more clearly presented empirical evidence is required to device recommendations in the style of “doubling model depth requires 4x more training data”.  For example, the \sqrt{D} dependence is not clearly obeyed by empirical results (Figure 3).

**Requested Changes:**

Questions

1.  Could you please help me understand, with Theorem 1, why “doubling model depth requires 4x more training data”, and “increasing kernel size from 3 to 12 doubles the data requirements”?

2. On Figure 2, could you please provide further definitions for the quantities plotted? For example, for the “theory” lines (dashed), are these evaluations of the Theorem 1 bounds, is C_0 replaced by N \beta(d), and what value of N is used (I assumed actual, not effective)?
For the “empirical” lines (solid), are these the empirical gaps or the ratio of empirical to theory? I am also a bit puzzled by the comment in the caption that “lower values indicate tighter bounds”.

3. Figure 4 is a bit confusing to me. I think it is because the y-axis has two separate scales for theory vs empirical, and it appears as if the theoretical bound is decaying faster.

In general, I think stronger/more clearly presented empirical evidence is required to device recommendations in the style of “doubling model depth requires 4x more training data”.  For example, the \sqrt{D} dependence is not clearly obeyed by empirical results (Figure 3).

---

> ### Author Response · Authors · 2025-10-07
>
> We thank you for the constructive feedback and positive assessment of our paper. We address the questions about empirical evidence below.
>
> **Clarifying the Theory-Practice Relationship.**
> Our theoretical bound provides a \emph{worst-case upper bound} across all $\beta$-mixing processes. Empirical performance depends on both estimation error (which our theory bounds) and approximation error (how well the architecture exploits temporal structure). For the AR(1) processes we study, temporal convolutional architectures achieve better approximation than worst-case analysis predicts, explaining why empirical results in Figure 3 show weaker depth dependence than the $\sqrt{D}$ bound.
>
> We will revise claims like ``doubling depth requires 4X more data'' to clearly state this as conservative theoretical guidance: ``our theoretical analysis suggests doubling depth requires approximately 4X more data to maintain worst-case guarantees, though architectures well-matched to temporal structure may require less.''
>
> **Addressing Specific Questions.**
>
> **Question 1: Why 4X more data?** From the bound term $R\sqrt{D/N}$, maintaining constant error when depth increases from $D$ to $2D$ requires $N' = 2N$. Doubling again to $4D$ requires $N'' = 4N$, hence ``quadrupling.'' This is conservative theoretical guidance; empirical requirements are lower when architectures exploit structure efficiently, as Figure 3 shows. We will clarify this calculation and emphasize its conservative nature.
>
> **Question 2: Figure 2 clarification.** Figure 2 plots the ratio of empirical generalization gap to theoretical bound on the y-axis (lower values indicate tighter bounds). The dashed lines at the top show how theoretical bounds scale with effective sample size for each $\rho$. The solid lines with markers show empirical gaps for each $\rho$. The dotted lines show power-law fits to the empirical data. The gray dashed reference line shows $N_{\text{eff}}^{-1/2}$ scaling. We will revise the caption to make these distinctions more explicit.
>
> **Question 3: Figure 4 dual axes.** Figure 4 (PhysioNet results) uses dual y-axes with different scales: the left axis (red, logarithmic scale from 100 to beyond 1000) shows theoretical bounds, while the right axis (blue, logarithmic scale from 0.0001 to 0.01) shows empirical gaps. The bounds are conservative upper bounds, not tight predictions, explaining why they appear on different scales. Both quantities decrease with $N$, with theoretical bounds showing $O(N^{-1/2})$ (dotted red reference line) and empirical gaps showing faster $N^{-0.79}$ convergence (dash-dot blue line). We will revise the caption to state: ``Theoretical bounds (left axis, red) and empirical gaps (right axis, blue) are plotted on different logarithmic scales because bounds are conservative upper bounds. Both decrease with $N$, with bounds showing the predicted $O(N^{-1/2})$ scaling and empirical gaps showing faster convergence due to structured regularities in physiological signals.''
>
> **Textual Revisions.**
> We will add explicit statements that our bounds establish sufficient rather than necessary conditions for generalization. We will add a discussion paragraph acknowledging the theory-practice gap in Figure 3, explaining that tighter bounds accounting for architectural inductive biases remain open problems, and positioning this as motivation for future work.
>
> **Why we think Contributions Remain Valid.**
> The theory-practice gap does not invalidate our contributions. Our bounds remain valid as conservative guarantees and represent the first architecture-aware analysis for dependent sequences. The fair-comparison methodology addresses systematic evaluation bias independently of bound tightness. The empirical findings motivate future theoretical work to better capture how architectures exploit temporal structure.

---

### Review · Reviewer_cYd2 · 2025-10-05

**Summary Of Contributions:**

This paper addresses the important problem of understanding generalization in deep temporal models, specifically Temporal Convolutional Networks (TCNs), trained on dependent (β-mixing) data. It proposes two main contributions:

1. Theoretical generalization bounds (Theorem 1) that claim to be "architecture-aware" and scale as $O(R\sqrt{D/N})$, suggesting only a square-root dependence on depth (D) and a linear dependence on the weight norm bound (R).
2. A "fair comparison methodology" for empirical evaluation, which controls for the effective sample size ($N_{eff}$) across sequences with different degrees of temporal dependence (ρ).

**Strengths:**
* The topic is highly relevant, as theoretical guarantees for deep temporal models are limited.
* The proposed "fair comparison methodology" is well-motivated and addresses a significant gap in how temporal models are typically evaluated. Controlling for $N_{eff}$ allows for a clearer understanding of how temporal structure affects learning, independent of information content.
* The empirical findings resulting from this methodology—suggesting that strong temporal dependence can enhance generalization—are interesting and counter-intuitive, warranting further study.

**Weaknesses:**
* The paper's main theoretical claims (Lemma 2, Proposition 1, and Theorem 1) suffer from fundamental mathematical errors and significant misinterpretations of prior work, particularly Golowich et al. (2018).
* The claimed $O(R\sqrt{D})$ scaling incorrectly omits the product of norms, hiding an exponential dependence on depth ($R^D$).
* The Online-to-PAC conversion framework used contains flaws regarding the necessary assumptions (convexity of loss) and the resulting concentration terms.

**Audience:**

No

**Audience Explanation:**

While the paper addresses a topic of high interest: generalization in deep temporal/sequential models, due to the fundamental errors in the paper, it is not of interest to the community and readers.

**Claims And Evidence:**

No

**Claims Explanation:**

The empirical contributions regarding the "fair comparison methodology" are supported by the evidence provided. However, the central theoretical claims of the paper—the architecture-aware generalization bounds presented in Lemma 2 and Theorem 1—are mathematically unsound and rely on critical misapplications of prior results.

The theoretical evidence provided is inaccurate due to the following major errors:

**1. Fundamental Misinterpretation of Golowich et al. (2018) and Omission of the Product of Norms (Lemma 2)**

The paper claims in Lemma 2 a Rademacher complexity bound for TCNs that scales as $O(R\sqrt{D})$. This bound is attributed to techniques from Golowich et al. (2018). This is incorrect.

Golowich et al. (Theorem 1) improved previous bounds that scaled as $O(2^D \prod_j^d M_F(j) )$ by removing the explicit $2^D$ factor, resulting in a bound scaling as $O(\sqrt{D} \prod_j^d M_F(j) )$ where $\prod_j^d M_F(j) $'s denote the bounds on **Frobenius norm** of weight matrices.  Crucially, the product of the norms, which could grow exponentially in depth remains. The present submission incorrectly omits this product of norms. This error is explicit in Appendix B.3 (Step 4, p. 30-31), where the authors state:

> "The key insight from Golowich et al. (2018) is that for networks with bounded weight norms, the depth dependence can be dramatically improved from $(R\sqrt{p})^{D}$ to $R\sqrt{pD}$."

This is a misrepresentation. Golowich et al. did not provide a mechanism to transform $R^D$ into $R\sqrt{D}$. Furthermore, the proof structure in Appendix B.3 attempts a standard Lipschitz composition argument (Steps 1-3), which inherently leads to exponential dependence, and then incorrectly applies Golowich's result post-hoc to eliminate this dependence. This approach is mathematically invalid. Golowich et al. in fact establish even a lower bound that has exponential dependence on depth.

Furthermore, the peeling technique used by Golowich et al. in (their Lemma 1) relies on **Frobenius norm**. The proof hinges on the structure of the Frobenius norm (sum of squared Euclidean norms of the rows). In contrast, the present submission constrains the network using the **$l_{2,1}$ norm** (p. 9). To apply Golowich's technique, the authors must prove that an equivalent lemma holds for the $l_{2,1}$ norm. This justification is missing. Therefore, even the $O(\sqrt{D} \cdot R^D)$ scaling is not substantiated for the hypothesis class defined in the paper.

**3. Flawed Online-to-Batch Conversion (Proposition 1)**

Proposition 1 attempts to establish a generalization bound for the average predictor $\bar{f} = \frac{1}{N}\sum_{t=1}^{N}h_{t}$ using online-to-batch conversion. This standard technique (e.g., Cesa-Bianchi & Lugosi, 2006) requires the loss function $l$ to be **convex** to apply Jensen's inequality. The paper only assumes the loss is bounded (p. 4).

This flaw manifests explicitly in the proof (Appendix B.2, Step 2, p. 29), where the authors state:
> "The term $\mathbb{E}[\tilde{L}_{1}]$ equals the true risk $\mathcal{L}(\bar{f})$ by stationarity."

This equality is incorrect. The equality fails due to the lack of convexity and the evolution of the algorithm over time. This invalidates Proposition 1.

**4. Incorrect Concentration Term (Proposition 1)**

The analysis uses $B = \lfloor N/(d+1)\rfloor$ blocks. The concentration inequality should depend on the number of approximately independent samples, $B$. In the proof (Appendix B.2, p. 29), the term is correctly identified as scaling with $1/\sqrt{B}$. However, in the final statement of Proposition 1 (p. 8), it is incorrectly presented as $\sqrt{\frac{\log(1/\delta)}{N}}$.

By replacing $B$ with $N$, the bound is artificially tighter. Since $d$ is chosen as $\Theta(\log N)$, the correct scaling should be approximately $\sqrt{(\log N)/N}$, not $\sqrt{1/N}$.

**Conclusion on Evidence:**
Due to these fundamental mathematical errors in the derivations of Lemma 2 and Proposition 1, the main theoretical result (Theorem 1) is invalid.

**Requested Changes:**

I welcome any clarification from authors on the questions/issues raised earlier. However, if my understanding and points raised are valid, given the severity of the theoretical issues, the authors might consider reframing the paper to focus entirely on the empirical methodology and findings. The "fair comparison methodology" can be an interesting contribution on its own.

---

> ### Author Response · Authors · 2025-10-07
>
> We thank you for your detailed technical critique. You correctly identify  four mathematical gaps in our original submission. We acknowledge these errors and provide corrections below. While the corrected bounds are weaker than originally claimed, we think the paper's core contributions remain substantive and merit publication.
>
> **Summary of Corrections.**
> We accept all four technical points raised. First, our Lemma 2 incorrectly dropped the product-of-layer-norms term. The correct bound retains $R=\prod_{\ell=1}^D M(\ell)$ and achieves $\sqrt{D}$ depth scaling, not elimination of the product term. Second, we used $\mathbb{E}[\tilde{L}_1] = L(\bar{f})$ without assuming convexity. This requires Jensen's inequality and thus demands convex, Lipschitz losses. Third, the concentration rate depends on the number of blocks $B = \lfloor N/(d+1)\rfloor$, yielding $O(\sqrt{\log N/N})$ under exponential mixing with $d^\star \asymp \log N$, not $O(1/\sqrt{N})$. Fourth, the step $R_N \le 2N\mathfrak{R}_N(\mathcal{F})$ lacked justification and is removed. We now present Proposition 1 as algorithm-agnostic in regret.
>
> **Corrected Main Result.**
> After corrections, our generalization bound becomes:
>
> $$\left|L(\bar{f})-\hat{L}_N(\bar{f})\right| \lesssim \frac{R_N}{N}+R\sqrt{\frac{D\log N}{N}}+\sqrt{\frac{\log(1/\delta)\log N}{N}} +o(1)$$
>
> where $R =\prod_{\ell=1}^D M(\ell)$ is the product of layer-wise weight norms, under convex Lipschitz losses with explicit norm control. This preserves the $\sqrt{D}$ architectural dependence together with the necessary norm-product factor.
>
> **Relationship to Golowich et al.\ (2018).**
> We do not claim to generalize or modify the Golowich et al.\ theorem itself. Their 2018 result provides an architecture-aware class complexity bound retaining the product-of-norms and improving depth dependence. We import that class bound and combine it with a separate, non-i.i.d.\ generalization analysis for $\beta$-mixing sequences via blocking, coupling, and block-level concentration. Thus, our generalization guarantee applies to dependent time series, but we are not extending the Golowich theorem, rather, we embed it within a mixing-aware learning framework. In this sense, our result extends the applicability of architecture-aware generalization analyses from i.i.d.\ samples to dependent sequences. The i.i.d.\ case emerges as a special case of our framework when $d=0$ and $B=N$.
>
> **What Remains Valid: Three Core Contributions.**
> The corrected work delivers three substantive contributions that we think justify publication.
>
> First, we provide the first principled bridge from architecture-aware complexity to dependent-sequence generalization via mixing theory. Our mixing-aware generalization guarantee for dependent time-series (via $\beta$-mixing, blocking/coupling, and block-level concentration) holds without i.i.d.\ assumptions, and the architecture-aware rate retains the functional depth dependence $\sqrt{D}$ together with the required norm product $R=\prod_{\ell=1}^D M(\ell)$. This makes prior i.i.d.\ architectural insights operational for time-series modeling.
>
> Second, we introduce a fair-comparison protocol that separates information content from temporal structure, addressing systematic biases in temporal learning evaluation. The corrected bound preserves actionable architectural guidance: depth dependence scales as $\sqrt{D}$ (alongside the norm product $R$), not exponentially. When we impose global norm constraints maintaining $R \le R_0$, the effective rate becomes $O(\sqrt{D\log N/N})$, yielding the practical insight that doubling network depth requires approximately quadrupling training data. Our fair-comparison methodology (matched norm budgets and effective-sample-size accounting) is intact and directly motivated by the corrected theory.
>
> Third, we provide empirical validation with explicit norm control. Our implementation calculates theoretical bounds using each model's actual measured weight norm after training. We record the final weight norm for every experiment in our results. Analysis of our synthetic experiments at fixed effective sample size $N_{\text{eff}}=2000$ and depth $D=4$ shows that weight norms remain nearly constant across different mixing coefficients, varying by less than 7\% (mean norms: $\rho=0.2$: 0.945, $\rho=0.4$: 0.957, $\rho=0.6$: 0.970, $\rho=0.8$: 0.927). This confirms that observed improvements do not arise from differential norm growth. The empirical finding that strongly dependent sequences ($\rho = 0.8$) achieve approximately 76\% smaller generalization gaps than weakly dependent sequences ($\rho = 0.2$) at this fixed effective sample size is obtained under these controlled conditions. We can provide our complete experimental code and results for verification.

---

> > ### Author Response · Authors · 2025-10-07
> >
> > **Why Empirical Claims Remain Valid Under Corrected Theory.**
> > The corrected theory decomposes generalization into estimation error (which our bounds capture) and approximation error (architecture-dependent but not analyzed in our theory). Our bounds quantify the estimation landscape under dependence while remaining agnostic about how well different architectures approximate the target function. For temporal data, architectures that exploit sequential structure achieve lower approximation error. In the study, the gain in approximation/predictability from temporal structure dominates the estimation penalty from dependence, yielding lower excess risk in practice. This is fully consistent with our corrected bounds, which favor architectures with controlled $R$ but do not preclude approximation benefits.
> >
> > Our experimental protocol controls for three critical factors: norm constraints through weight decay and explicit projection when norms exceed bounds; effective sample size calculated as $N = N_{\text{eff}} \cdot (1+\rho)/(1-\rho)$ to maintain constant information content across different mixing strengths; and architectural consistency with fixed depth $D$, kernel size $p$, and hidden dimensions across comparisons. Under these controls, the observed trend: that temporal structure improves generalization when architectures are well-matched, aligns with the corrected theoretical picture. The theory bounds estimation error; experiments show that approximation benefits can outweigh estimation costs.
> >
> > **What We No Longer Claim.**
> > We retract four claims from the original submission. First, we do not claim a depth bound eliminating the product-of-norms; it properly appears as $R\sqrt{D}$. Second, we do not claim $O(1/\sqrt{N})$ concentration under mixing; the correct rate is $O(\sqrt{\log N/N})$. Third, we do not claim the unqualified equality $\mathbb{E}[\tilde{L}_1] = L(\bar{f})$ without convexity assumptions. Fourth, we do not claim the direct link $R_N \le 2N\mathfrak{R}_N(\mathcal{F})$ without additional justification. While the corrected rate is slightly weaker (extra $\sqrt{\log N}$ factor and explicit $R$), the functional dependence on depth remains $\sqrt{D}$ (up to logs), and the directional trend in the bounds aligns with our experiments: for fixed depth $D$ and matched norm budgets, models that better exploit temporal structure achieve lower generalization error.
> >
> > **Required Revisions.**
> > We will make five key revisions to the manuscript. First, we will restate Lemma 2 to include $R = \prod_\ell M(\ell)$ explicitly, clarifying that we achieve $\sqrt{D}$ scaling alongside this factor (as in Golowich et al.). Second, we will add convexity and Lipschitz requirements to Proposition 1, replacing the equality with Jensen's inequality. Third, we will correct concentration rates to reflect block-level dependence: $O(\sqrt{\log N/N})$ under exponential mixing. Fourth, we will remove the unjustified regret-Rademacher claim and present Proposition 1 as algorithm-agnostic with regret instantiated via standard OCO bounds. Fifth, we will clarify throughout that we \emph{apply} Golowich et al.'s result within a mixing-aware framework rather than extending or modifying their theorem, and will note explicitly that the i.i.d.\ setting is a special case of our analysis.
> >
> > **Conclusion.**
> > Even with the corrected bounds, the paper delivers a principled, mixing-aware extension of architecture-aware generalization to dependent time series, preserves the $\sqrt{D}$ architectural message under explicit norm control, and empirically validates the predicted trends under matched budgets with quantitative evidence of controlled norms. WE think these contributions enable researchers to apply depth-scaling insights to time series problems with proper theoretical justification and evaluation methodology, making a solid case for publication.

---

> ### Comment · Reviewer_cYd2 · 2025-10-10
>
> I appreciate the authors honesty and acknowledgment of the mathematical gaps.
>
> However, the proposed corrections represent a fundamental change in the paper's core contribution. The original's central theoretical claim, now retracted, was a significant theoretical advance. In contrast, the corrected result is a straightforward synthesis of known techniques—combining an existing Rademacher complexity bound with a standard blocking method for dependent data.  Furthermore, the scale of these changes is too substantial for a rebuttal. A new main theorem with different assumptions and much weaker guarantees constitutes a different paper in substance. It is, in effect, a new submission that requires a fresh and independent review cycle.
>
> Finally, the nature of the original errors is concerning. Misinterpreting the main claims of literature and overlooking basic assumptions points to a lack of diligence & rigor that goes beyond typical calculation or derivation errors. I urge the authors to apply greater rigor in future theoretical work. For these reasons, my recommendation remains to reject.

---

### Decision · Action_Editor_Vi8v · 2025-11-07

**Recommendation:** Reject

**Audience:**

No

**Audience Explanation:**

Although the reviewers agreed that the original message of the paper was interesting to TMLR's audience, the corrected results are much weaker than the originally proposed and mainly amount to incremental changes of lower interest for the community.

**Claims And Evidence:**

No

**Claims Explanation:**

The reviewers spotted significant errors in the original proofs that were acknowledged by the authors during the rebuttal period. Fixing those errors required deviating significantly from the original theory. The reviewers agreed that the differences are so significant that they amount to a new submission which would require a new round of reviews.

**Resubmission Of Major Revision:**

The authors may consider submitting a major revision at a later time.